# EvoSBDD: Latent Evolution for Accurate and Efficient Structure-Based Drug Design

**Danny Reidenbach** [1]

## Abstract

Structure-based Drug Design (SBDD), the task of designing 3D molecules (ligands) to bind with a target protein pocket, is a fundamental task in drug discovery. Recent geometric deep learning methods for SBDD fail to accurately generate valid docked structures without relying on physics-based post-processing (*i.e.* AutoDock Vina (ADV) redocking), which resamples all the important geometric qualities of the molecule. To best target the generation of accurate docked molecules, we reframe SBDD as a 1D-controlled latent generation problem. Instead of relying on 3D structures, we introduce EvoSBDD which performs a simple evolutionary algorithm in a pretrained 1D latent space using an ADV redocking oracle. Without 3D structure information or additional training on protein-ligand complexes as required by prior methods, EvoSBDD attains a state-of-the-art success rate of 86.4%, an average binding affinity of -10.27 kcal/mol, and demonstrates speed improvements up to 25.6x compared to the prior best method. EvoSBDD is the first method to maintain 100% generated molecule validity, novelty, and uniqueness and also excels in real-world off-target(s) binding prevention.

## 1. Introduction

Designing effective drug-like molecules for diverse protein targets is a pivotal challenge in drug discovery. In recent years Structure-based Drug Design (SBDD), which aims to generate 3D ligands conditioned on target protein pockets, has been a significant focus of geometric deep learning. Majority of recent methods follow a similar structure-based pipeline where given a ligand and target protein with 3D atomic positions and discrete atom/residue types, they train an equivariant diffusion model based on Hoogeboom

et al. (2022) to generate binding molecules. While primarily successful, diffusion-based methods do not fully outperform prior auto-regressive SBDD methods (Peng et al., 2022) when it comes to the desired chemical properties, and often struggle with structural connectivity and 2D validity (Schneuing et al., 2022; Guan et al., 2023b), even with explicit validity sampling guidance.

As the base architectures for all current diffusion-based SBDD methods are nearly identical, the subsequent improvements from one to another are overshadowed by the increase in computational cost. In its current form, the structure-based pipeline is misaligned with the primary metric of current SBDD evaluation, the binding affinity of the generated 3D ligand and target protein calculated by AutoDock Vina (ADV) (Eberhardt et al., 2021) redocking. Upon receiving a generated 3D ligand, ADV redocking performs rotation, translation, and re-sampling of all the torsion angles. Thus, this alters the geometry of the molecule significantly, making it largely independent of the initially generated structure. There are ADV-score-only metrics that assess the accuracy of the generated structure without redocking. However, existing diffusion-based methods struggle to match the SBDD baseline (Francoeur et al., 2020) and thus are not emphasized as much. Furthermore, there exist many cases where poor score-only results possess strong redocked results, which demonstrates how most of the heavy lifting is being done by ADV (Guan et al., 2023a). Our work tackles the following question: To generate molecules exhibiting the strongest best-case binding affinity (*i.e.* through redocking that re-samples the molecular geometry), is it necessary for our model to learn 3D structures?

In this work, we propose EvoSBDD, an evolutionary pipeline that, uses a naive black-box optimization (BBO) over the latent space of a pretrained molecule autoencoder (LLM or GNN) for efficient and accurate SBDD. As SBDD is evaluated primarily on the ADV redocking score and array of chemical properties that do not depend on ligand 3D structure, we can remove the need to model 3D molecule structures and the target protein pocket entirely. EvoSBDD generates 2D molecule graphs via 1D SMILES embeddings, and uses a random RDKit 3D conformer as input to an efficient ADV docking oracle to produce an

[1]NVIDIA. Correspondence to: Danny Reidenbach <dreidenbach@nvidia.com>.

*Proceedings of the 41st International Conference on Machine Learning*, Vienna, Austria. PMLR 235, 2024. Copyright 2024 by the author(s).

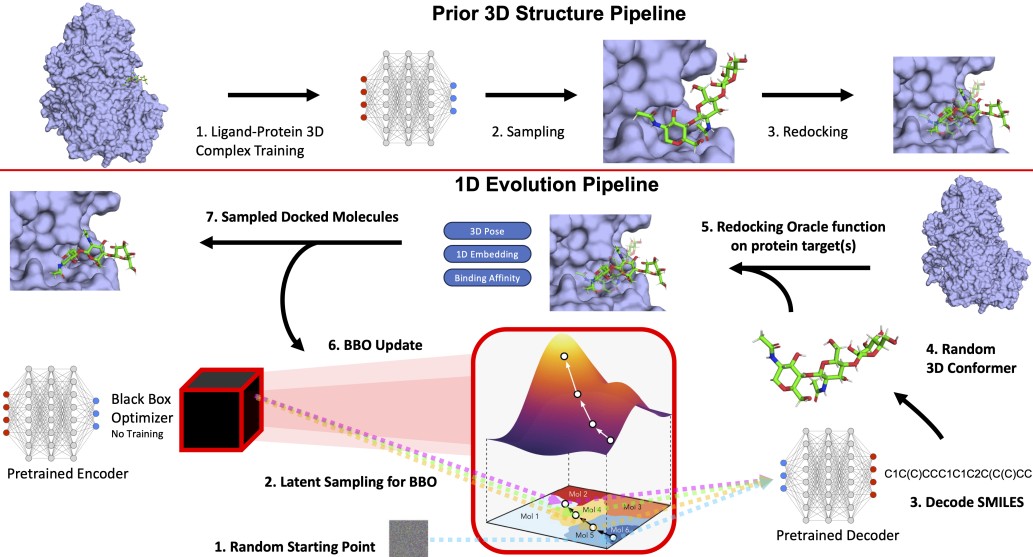

Figure 1: **Evolution Pipeline**: Controlled latent space optimization for EvoSBDD[1]. Leverage docking directly via BBO rather than learning an initial structure that is adjusted (rototranslation + torsion angle resampling) with little dependence on input structure for final binding affinity calculations.

accurate docked pose. In contrast to prior structure-based methods that model discrete atom types and continuous atom positions which rely on existing software to predict bonds based on inter-atomic distances, EvoSBDD isolates its success to 2D molecule (atom types and valid bonds) generation. Overall, we demonstrate how translating from 3D structure generation to 1D controlled latent generation results in vastly better binding molecules with higher desired property scores ( Fig. 1).

Our main contributions are as follows:

- EvoSBDD obtains state-of-the-art results across existing CrossDocked2020 (Francoeur et al., 2020) and PoseCheck benchmarks (Harris et al., 2023). EvoSBDD achieves a success rate of 86.4% with an average binding affinity of -10.27 kcal/mol *vs* 52.5% and -8.98 kcal/mol of prior best method.

- EvoSBDD is the first method to maintain 100% sampled molecule validity, novelty, and uniqueness with strong diversity across all target protein pockets of the CrossDocked2020 test set.

- EvoSBDD is the first method to offer a trade-off between accuracy and inference time. 25.6x faster at the lowest accuracy (+21% success) and 1.5x faster at the highest accuracy (+33.9% success) compared to prior best method. Furthermore, EvoSBDD's BBO can be naively parallelized to achieve speedups of 205x.

- We present a novel multi-target specificity benchmark for SBDD, mimicking real-world drug design scenarios. EvoSBDD simultaneously optimizes for one or

more binding sites while avoiding off-target site(s)—an achievement not possible with previous structure-based methods designed for single protein structures.

EvoSBDD demonstrates that if the goal is to generate the best docking scores, with desirable chemical properties, 1D latent search outperforms existing 3D structure methods.

## 2. Methods

**Overview** We refer to EvoSBDD as the process of applying a simple evolutionary algorithm with protein docking oracle functions to a latent space pretrained on 2D molecules via 1D SMILES embeddings (no 3D coordinates or protein information) for structure-based drug discovery. By taking advantage of the high molecular validity and speed of 1D latent-based molecule generation, we efficiently optimize for various 3D protein targets without additional training or providing 3D information directly to the model. Furthermore, rather than learning to encode the target protein information, we frame the problem as a conditional molecule generation problem to enable the use of pretrained molecule autoencoders with zero explicit protein information. In contrast to prior the 3D-structure pipeline, which encodes the protein target structure and learns to generate the ligand and its pose, our 1D pipeline more efficiently generates docked 3D structures with desirable chemical properties ( Fig. 1).

Overall, EvoSBDD is designed for real-world practitioner usage. We focus on inference time and existing methods of measurable molecule quality as the central normalizing criteria when comparing different methods, as those are

---

[1]Latent space diagram from Gómez-Bombarelli et al. (2018).

crucial when deciding which tools to use for real-world drug design. We emphasize that by removing the structure modeling paradigm of prior SBDD methods (*i.e.* no coordinate input or structure generation) that ADV overrides in its redocking procedure, we create a simple, efficient, and accurate SBDD method. To our knowledge, EvoSBDD is the first method capable of using the full ADV docking pipeline as prior structure-based methods cannot use gradient-free structure altering mechanisms as it would distort the training objective and would be too slow to use in practice. EvoSBDD's direct latent optimization yields strong performance across various chemical properties unknown to the optimization procedure, whereas some prior methods require explicit supervision. We also demonstrate how EvoSBDD's oracle function can include any combination of desired chemical properties and protein target(s) docking.

Furthermore, EvoSBDD demonstrates that generative SBDD success is greatly tied to the actual molecule/bond identity as the 3D pose can be constructed with existing efficient physics-based tools. Blindly learning conditional structures is not enough for highly accurate SBDD, and even proxy-based or ADV score-only optimizations fall short in existing benchmarks (Shen et al., 2023; Zhou et al., 2024).

**Molecular Latent Space** As EvoSBDD generates molecules by perturbing pretrained latent representations, here we describe the architectures and pretraining procedures of the models experimented with. We evaluated EvoSBDD on two pretrained molecular probabilistic autoencoders. The first being that of MoFlow (Zang & Wang, 2020), a normalizing flow-based graph generative model, which we denote as EvoSBDD-GNN. MoFlow has 141M parameters and was pretrained on ZINC-250K (Irwin et al., 2012) with a latent space dimension of 6800. We also use a chemical LLM in EvoSBDD . Specifically, we use MolMIM (Reidenbach et al., 2022), a probabilistic autoencoder that learns an informative and clustered latent space by optimizing for mutual information rather than KL divergence term of a traditional VAE. The model's encoder is a Perceiver (a fixed output length transformer) and the decoder is a seq2seq transformer, identical to the one used in BART (Lewis et al., 2020). MolMIM has a latent space dimension of 512, 65.2M parameters, and was trained on 730M molecules of ZINC-15 (Sterling & Irwin, 2015).

**Evolutionary Algorithm** We employ a simple evolutionary algorithm to generate molecules directly for SBDD directed by an ADV docking oracle function. Specifically, we use Covariance Matrix Adaptation Evolution Strategy (CMA-ES, Algorithm 2) (Hansen, 2006), a greedy, gradient-free (*i.e.*, $0^{th}$ order optimization) evolutionary search algorithm that maximizes a black-box reward function directly in the latent space. CMA-ES maintains a distribution over

candidate solutions, adapting both the mean and covariance matrix of the distribution to efficiently explore and exploit the search space for finding optimal solutions. We chose CMA-ES as it is easy to use albeit a naive algorithm and an overall weak baseline for novel optimization methods (Yang et al., 2021). Furthermore, CMA-ES can always be replaced with more rigorous BBO algorithms like LA-MCTS (Yang et al., 2021), but we purposely chose to start with a weak optimizer to best evaluate the overall EvoSBDD 1D pipeline.

We use four hyperparameters for CMA-ES: iterations, population size (ps, optimization batch size), standard deviation (sigma, controls the step size in the search space), and restarts (R, number of trials). For example, with a population size of 20, with 5 iterations, we generate a total of 100 latent points. We can do this for 3 restarts to have a total number of 300 generated molecules. Reidenbach et al. (2023) has shown that CMA-ES performs better when breaking the total desired samples into independent restarts, which also decreases compute costs via the ability to run restarts in parallel. We use the identity matrix for the initial covariance matrix and initial step size of 0.25 and 1 for the GNN and LLM latent spaces, respectively. For more details on CMA-ES( Algorithm 2) and its default parameters, see §B.

**Oracle Function** Algorithm 1 provides pseudocode for generating $L$ molecules for a desired protein pocket which depends on its AutoDock Vina (ADV) docking oracle function. Specifically for each reference molecule-protein pair in the test set, we construct an ADV oracle function centered at the center of the desired protein pocket. ADV is then allowed to explore a cube with length 20 angstroms around the center to find the best pose using its rule-based evaluation detailed in §C.2. Furthermore, as base ADV is too slow to run at an efficient scale, we equip EvoSBDD with UniDock (Yu et al., 2023), which is a GPU-accelerated ADV scoring function. UniDock enables docking K ligands in parallel to the same protein target, which fits nicely with the BBO paradigm. For more details about UniDock and how it defers from ADV please see §C.3.

## 3. Experiments

**Data** We utilized the CrossDocked2020 dataset (Francoeur et al., 2020) to evaluate EvoSBDD . As done in Ragoza et al. (2022); Peng et al. (2022); Guan et al. (2023a;b); Zhou et al. (2024), we generate 100 ligands for each of the same 100 test set proteins for evaluation. We note that unlike prior work we do not train on CrossDocked or any 3D structure or protein information. The only protein or 3D information the model can see is through the scalar oracle score during the BBO.

**Metrics** Following Guan et al. (2023a) we evaluate the generated molecules on protein binding affinity and various

Table 1: Structure-based Drug Discovery Benchmarks. $\alpha = 1$ and $\sigma = 0$ unless specified. See Tab. 3 for full results.

| | | Validity (↑) | Vina Dock (↓) | | High Affinity (↑) | | QED (↑) | | SA (↑) | | Diversity (↑) | | Lipinski (↑) | Success Rate (↑) | Time (↓) |
| | | Avg. | Avg. | Med. | Avg. | Med. | Avg. | Med. | Avg. | Med. | Avg. | Med. | Avg. ± Std. | Avg. | Gen + Score. |
|---|---|---|---|---|---|---|---|---|---|---|---|---|---|---|---|
| | Reference | 100% | -7.45 | -7.26 | - | - | 0.48 | 0.47 | 0.73 | 0.74 | - | - | 4.34 ± 1.14 | 25.0% | 300 |
| **Generative** | liGAN | - | -6.33 | -6.20 | 21.1% | 11.1% | 0.39 | 0.39 | 0.59 | 0.57 | 0.66 | 0.67 | - | 3.9% | - |
| | GraphBP | - | -4.80 | -4.70 | 14.2% | 6.7% | 0.43 | 0.45 | 0.49 | 0.48 | **0.79** | **0.78** | 4.83 ± 0.37 | 0.1% | 310 |
| | AR | 92.95% | -6.75 | -6.62 | 37.9% | 31.0% | 0.51 | 0.50 | 0.63 | 0.63 | 0.70 | 0.70 | 4.78 ± 0.51 | 7.1% | 19959 |
| | Pocket2Mol | 98.31% | -7.15 | -6.79 | 48.4% | 51.0% | 0.56 | 0.57 | 0.74 | 0.75 | 0.69 | 0.71 | 4.93 ± 0.27 | 24.4% | 2804 |
| | TargetDiff | 90.35% | -7.80 | -7.91 | 58.1% | 59.1% | 0.48 | 0.48 | 0.58 | 0.58 | 0.72 | 0.71 | 4.59 ± 0.83 | 10.5% | 3728 |
| | DiffSBDD | 85.01% | -8.03 | -7.74 | 55.3% | 56.6% | 0.47 | 0.47 | 0.55 | 0.56 | 0.76 | 0.76 | 4.70 ± 0.64 | 6.0% | 460 |
| | DecompDiff | 71.96% | -8.39 | -8.43 | 64.4% | 71.0% | 0.45 | 0.43 | 0.61 | 0.60 | 0.68 | 0.68 | 4.29 ± 0.97 | 24.5% | 6489 |
| **Optimization** | TacoGFN | 99.27% | -7.41 | -7.50 | 58.9% | 59.0% | **0.68** | **0.72** | 0.78 | 0.79 | 0.65 | 0.65 | 4.94 ± 0.24 | 29.9% | 303 |
| | TacoGFN-AL | 99.28% | -7.68 | -7.70 | 64.3% | 64.0% | 0.64 | 0.66 | **0.81** | **0.82** | 0.66 | 0.66 | 4.93 ± 0.25 | 36.6% | 303 |
| | RGA | - | -8.01 | -8.17 | 64.4% | 89.3% | 0.57 | 0.57 | 0.71 | 0.73 | 0.41 | 0.41 | - | 46.2% | - |
| | TargetDiff+Opt | - | -8.30 | -8.15 | 71.5% | 95.9% | 0.66 | 0.68 | 0.68 | 0.67 | 0.31 | 0.30 | - | 25.8% | >3728 |
| | DecompOpt | - | -8.98 | -9.01 | 73.5% | 93.3% | 0.48 | 0.45 | 0.65 | 0.65 | 0.60 | 0.61 | - | 52.5% | 9241 |
| **Ours** | **EvoSBDD** ($\alpha = 0, \sigma = 1$, 8R) | **100%** | **-9.09** | **-9.20** | 82.1% | 100% | 0.65 | 0.67 | 0.78 | 0.79 | 0.65 | 0.66 | **4.96 ± 0.21** | **73.5%** | 360 |
| | **EvoSBDD** ($\sigma = 1.3$, 140R) | **100%** | **-10.27** | **-10.36** | 96.5% | 100% | 0.53 | 0.52 | 0.75 | 0.77 | 0.63 | 0.63 | 4.84 ± 0.44 | **78.8%** | 6300 |
| | **EvoSBDD** ($\alpha = 0, \sigma = 1$, 140R) | **100%** | **-10.14** | **-10.27** | 94.4% | 100% | 0.59 | 0.59 | 0.77 | 0.77 | 0.62 | 0.62 | 4.91 ± 0.29 | **86.4%** | 6300 |

important chemical properties: (1) **Validity** is the percentage of connected and valid generated molecules determined by RDKit that also have non-positive and thus plausible binding free energies. (2) **Vina Dock** approximates binding affinity between molecules and their target pockets (Eberhardt et al., 2021); (3) **High Affinity** measures the percentage of generated molecules with higher affinity than the reference molecule; (4) **QED** measures the drug-likeness of a molecule based on its physicochemical properties and structural features; (5) **Synthetic Accessibility (SA)** estimates how easily the molecule can be synthesized and is normalized to a range of 0 to 1 via (10 - SA)/9 (Ertl & Schuffenhauer, 2009). (6) **Diversity** is the average pairwise fingerprint dissimilarity between generated molecules for each target; (7) **Lipinski** measures the number of rules satisfied in Lipinski rule five (Lipinski et al., 1997); (8) **Success Rate** is the percentage of molecules which pass certain criteria (QED > 0.25, SA > 0.59, Vina Dock < -8.18) for comprehensive evaluation; (9) **Inference Time** is the average time in seconds to generate 100 molecules and dock them to one target pocket.

The ADV success threshold ($< -8.18$ kcal/mol) corresponds to $1\mu$M binding affinity, which is a common requirement for a potential drug candidate in practical drug discovery; the QED and SA thresholds are calculated as the 10th percentile of DrugCentral (Ursu et al. (2016) a database of up-to-date drugs and pharmaceuticals), to reflect the latest drug property distribution. These thresholds are also used by Guan et al. (2023b); Long et al. (2022); Zhou et al. (2024). For a further discussion on the relationship between ADV docking score and drug binding kinetics please see §E.

**Baselines** Following Zhou et al. (2024) we separate existing SBDD methods by the perspective of generation and

optimization. Generative methods sample molecules for a given protein target whereas optimization methods guide the generation based on certain criteria. Generative methods we compare against include: liGAN (Ragoza et al., 2022), AR (Luo et al., 2021), Pocket2Mol (Peng et al., 2022), GraphBP (Liu et al., 2022), TargetDiff (Guan et al., 2023a), DiffSBDD (Schneuing et al., 2022), and DecompDiff (Guan et al., 2023b). Optimization methods we compare against include TacoGFN (Shen et al., 2023), RGA (Gao et al., 2022), TargetDiff+Opt and DecompOpt (Zhou et al., 2024) (which are akin to greedy optimization over diffusion model-like samples). These methods exhibit direct supervision over the optimization of binding affinity and key chemical properties. Unlike prior optimization-based methods, EvoSBDD only needs the ADV docking function, no validity, clash, or explicitly chemical property guidance is required. We note EvoSBDD's oracle function can be supplemented with desired chemical properties in Appendix §D.

**CrossDocked Evaluation** Compared to prior generative and optimization-based methods, EvoSBDD achieves state-of-the-art results for binding affinity, high affinity, Lipinski, and success rate while maintaining strong results for all chemical properties ( Tab. 1). EvoSBDD is the only method to not be deficient in any one area, offering the best balance of strong docking, property performance, and efficiency. Unlike prior BBO methods for molecule design that rely on a seed molecule to initialize the optimization, we found that removing this reference dependence by adding large amounts of Gaussian noise to the initial CMA-ES mean ($\sigma = 1.3$) and removing the reference molecule from the optimization entirely ($\alpha = 0, \sigma = 1$ Algorithm 1) significantly improves nearly all metrics. This demonstrates that like the diffusion models, EvoSBDD can generate drug-like molecules with high binding affinity from pure noise with-

Table 2: Validity Normalized Binding Affinity and CrossDocked2020 Success Rate.

| | Validity (↑) | Success Rate (↑) | | Vina Dock (↓) | | | |
| --- | --- | --- | --- | --- | --- | --- | --- |
| | | No Normalization | Normalized Validity | No Normalization | | Normalized Validity | |
| | | | | Avg. | Med. | Avg. | Med. |
| Reference | 100% | 25.0% | 25.0% | -7.45 | -7.26 | -7.45 | -7.26 |
| Toy Example | 1% | 100% | 1% | -8.30 | -8.30 | -0.08 | -0.08 |
| AR | 92.95% | 7.1% | 6.6% | -6.75 | -6.62 | -6.27 | -6.42 |
| Pocket2Mol | 98.31% | 24.4% | 24.0% | -7.15 | -6.79 | -7.03 | -6.75 |
| TargetDiff | 90.35% | 10.5% | 9.5% | -7.80 | -7.91 | -7.04 | -7.62 |
| DiffSBDD | 85.01% | 6.0% | 5.1% | -8.03 | -7.74 | -6.82 | -7.14 |
| DecompDiff | 71.96% | 24.5% | 17.6% | -8.39 | -8.43 | -6.03 | -6.02 |
| EvoSBDD ($\alpha = 0, \sigma = 1$, 140R) | **100%** | **86.4%** | **86.4%** | **-10.14** | **-10.27** | **-10.14** | **-10.27** |

out the need for a reference or seed molecule as required by other BBO methods. We emphasize EvoSBDD only optimizes for binding whereas prior optimization methods optimize for other properties such as QED and SA. Please see Tab. 5 for detailed ablations including seed-conditioned SBDD, various CMA-ES parameterization, LLM, GNN latent spaces, and multi-objective oracle function designs. We also provide an analysis of the impact of reference binding affinity on seed-conditioned generative success rate in Appendix §D.

**CrossDocked Efficiency** EvoSBDD uses UniDock (single A6000 gpu) unless ADV is specified. For evaluation consistency, as UniDock overestimates scores by 0.14-0.34 kcal/mol on average, the poses are re-scored with Vina (score only, no re-docking, so the structure remains the same) to ensure a fair comparison. For further discussion on UniDock, please see §C.3. The prior method results in Tab. 1 were calculated with their published generated molecules if available or taken from their publication. We specify the number of BBO restarts (R) for EvoSBDD and note each restart can be run in parallel in 45 seconds. We report the serial time, representing the most compute-constrained setting for fairness.

We emphasize that the majority of our speed-up is due to our iterative BBO design. For all prior 3D structure methods, replacing ADV redocking with UniDock would reduce inference time by 250 seconds, which is an improvement of 0.3% of the prior best method. Furthermore, as demonstrated in Tab. 3 we still achieve SOTA results in significantly less time without UniDock. We note that even if diffusion models wanted to leverage UniDock for sample guidance it would be too slow as it would require 500-1000 sequential calls due to the SDE solving step. In addition, ADV structural perturbations may cause a distribution shift as the geometric changes by the docking simulation (largely independent of the initial structure) would not have a gradient signal back to the diffusion dynamics. Overall, EvoSBDD offers a balance between generation speed, accuracy, and computational resources while maintaining strong performance on competitive SBDD benchmarks.

**Impact of Validity** Like 2D ML methods for molecule generation (Jin et al., 2020), 3D SBDD methods must possess strong validity of their generated molecules to be usable for downstream drug design. We added validity to Tab. 1, which is the percent of connected valid molecules (those that have physical binding affinity $\leq 0$ and can be processed by RDKit) of the total 10,000 sampled (100 molecules per 100 proteins). As prior methods do not guarantee generating 100 ligands for each test protein pocket, the average results of Tab. 1 are not taken over the same number of samples. To more fairly compare different SBDD methods, we normalize the docking scores in Tab. 2. For each method, whenever it fails to generate a valid molecule, it receives a docking score of 0 (rather than include its potential true value $> 0$, which would majorly skew results). We only do this for docking scores as normalization would decrease other metrics in a non-meaningful way (QED and SA would be artificially low). For further experiments that analyze EvoSBDD's (1) novelty, similarity, and uniqueness (Tab. 6), (2) PoseCheck benchmark results (§G), and (3) newly introduced multi-target specificity optimization benchmarks (§F), please see the Appendix.

## 4. Conclusions

Given a protein target of interest, EvoSBDD efficiently generates state-of-the-art molecules with strong binding affinity, while also possessing desirable chemical properties including drug-likeness and synthesizability. Instead of using an expensive 3D diffusion model, or slow 3D autoregressive model, EvoSBDD does all generation using a pretrained 1D latent representation that does not depend on any 3D structure or protein information. EvoSBDD demonstrates that when optimizing for AutoDock Vina (ADV) redocking and other non-structural properties like QED, significant success can be seen by only focusing on the discrete atom types and interatomic bonds, leaving all structure generation to our ADV or UniDock oracle. Beyond existing Cross-Docked2020 benchmarks, EvoSBDD excels at PoseCheck evaluations as well as newly introduced real-world multi-protein target design problems.

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

# A. Background

**Effective Molecule Representations**   Representation learning for biological data has shown useful for a multitude of tasks. Large Language Models (LLM) for proteins (ESM2 (Lin et al., 2023)) and small molecules (MolMIM (Reidenbach et al., 2022) and MolFormer-XL (Ross et al., 2022)) learn meaningful representations that can be leveraged in generative and predictive tasks. LLMs for proteins have a direct relationship due to the scientific importance of a protein's amino acid sequence whereas molecules, as they are inherently graphical by nature, have many equivalent ways to serialize them in a sequence format (Krenn et al., 2022). As a result, GNN autoencoders such as MoFlow (Zang & Wang, 2020) target molecule graphical structure to learn an effective representation. We evaluate EvoSBDD over LLM (MolMIM) and GNN (MoFlow) representations to leverage the rich information of existing off-the-shelf pretrained molecule models.

**Optimization Methods**   Once there exists an information-rich latent representation there are many optimization techniques at one's disposal for generative modeling. Prior molecule variational autoencoders have trained property predictors from the latent representation to use predictor gradient steps to move in the latent space to locate optimized molecules (Gómez-Bombarelli et al., 2018). Outside of gradient steps that can be rendered ineffective by non-smooth representations, there have been several reinforcement learning (RL) approaches to learn the best way to move in the latent space (Kearnes et al., 2019). While RL is an effective tool in many scenarios, it can be difficult to train in new situations, especially for complex reward functions.

As a result, methods turn to Black-Box Optimization (BBO), which is designed to optimize objective functions where the underlying mathematical form or structure is unknown or highly complex. In these algorithms, the objective function is treated as a "black box" that takes input parameters and produces an output (the objective value) without revealing the internal details of the function. In cases like molecular docking, as AutoDock Vina (ADV) relies on a highly complex and stochastic physics-based algorithm(see §C), we turn to BBO to better explore the space of drug-like docking molecules. We note BBO has been used in other generative molecule tasks like synthesis tree design (Reidenbach et al., 2023) and molecule optimization (Iwata et al., 2023).

**Structure-based Drug Design**   Several approaches for SBDD have been explored. liGAN (Ragoza et al., 2022) is a 3D CNN-based conditional VAE model that generates ligand molecules in atomic density grids. AR (Luo et al., 2021), Pocket2Mol (Peng et al., 2022) and GraphBP (Liu et al., 2022) are GNN-based methods that generate 3D molecules atom by atom in an autoregressive manner. TargetDiff (Guan et al., 2023a), DiffBP (Lin et al., 2022) and DiffSBDD (Schneuing et al., 2022) are equivariant diffusion-based methods based on Hoogeboom et al. (2022) for continuous position and discrete atom type generation. DecompDiff (Guan et al., 2023b) extends the diffusion framework with decomposed priors, additional bond type diffusion, and validity guidance. RGA (Gao et al., 2022) uses a genetic algorithm with policy networks while TacoGFN (Shen et al., 2023) uses a conditional GFlowNet for SBDD optimization. DecompOpt (Zhou et al., 2024) combines a pre-trained equivariant decomposed and conditional diffusion model (*i.e.* like DecompDiff) to extract a molecular grammar with a greedy iterative optimization algorithm to improve the desired properties. We note several optimization methods optimize for ADV score, QED, and Synthetic Accessibility (SA) at the same time whereas EvoSBDD outperforms prior methods on all properties only optimizing for ADV redocking.

**Binding Affinity Prediction**   AutoDock Vina (ADV) and similar variations (smina, gnina, qvina) all exist to predict the binding affinity between a target ligand and protein via a score-based gradient-free optimization (Eberhardt et al., 2021). These simulation programs offer a nice balance between reasonable accuracy and speed as more accurate methods are orders of magnitude more expensive. For more details about how ADV works and its underlying score function see §C. As SBDD is designed to achieve the best binding affinity, these physics-based docking tools have become a focal point for molecule generation benchmarking. Specifically, ADV supports three modes of operation. (1) **score-only** where given a ligand its binding affinity for a target protein is evaluated at the exact pose given. (2) **minimization** where the ligand undergoes a local BFGS energy optimization where the atomic positions are adjusted to reach an energy minima. (3) **redocking** which involves a series of global and local optimizations where ADV random initializes several ligand poses via torsion angle sampling followed by local energy minimization. As redocking provides the most accurate binding affinity predictions it is commonly focused on as the central SBDD metric.

# B. CMA-ES Details

---

**Algorithm 1** Structure-based Drug Design 1D Evolution

---

**Input:** Protein $P$, Reference molecule $m$ and SMILES $s$, Noise perturbation $\sigma$, Number of restarts $R$, Pretrained model $M$, oracle function(s) $F$ that returns binding affinity value, desired number of generated molecules $L$, binary flag for reference molecule usage $\alpha$

**Output:** Set of generated 3D molecules with docked poses $\mathbf{X}^*$

**Input:** Encode SMILES $s$ with model $M$ to obtain initial embedding $\mathbf{m}_0$

**Initialize:** $\mathbf{X}^* \leftarrow \emptyset$ {Each restart can be run in parallel}

**for** $r = 1$ **to** $R$ **do**

    **Noisy Embedding:** $\mathbf{m}_r \leftarrow \alpha \cdot \mathbf{m}_0 + \sigma \cdot \mathcal{N}(\mathbf{0}, \mathbf{I})$

    **Optimization with CMA-ES:** Apply CMA-ES with initial mean $\mathbf{m}_r$ and decode with $\mathbf{M}$ to get molecule solution set $\mathbf{Y}$

    **Decode and Dock Protein:** $\mathbf{V} \leftarrow F(\mathbf{Y}, P)$

    **Update CMA-ES with scores:** Use $\mathbf{V}$ to update CMA-ES parameters {See Alg. 2}

    **Store Result:** $\mathbf{X}^* \leftarrow (\mathbf{Y}, \mathbf{V})$

**end for**

**Sort Results:** Sort $\mathbf{X}^*$ by its score component

**return** $\mathbf{X}^*[: L]$ {Top L molecules}

---

**Algorithm 2** CMA-ES: Covariance Matrix Adaptation Evolution Strategy

---

1: **Input:** Objective function $f$, population size $\lambda$, initial mean $\mathbf{m}$, initial covariance matrix $\mathbf{C}$, initial step size $\sigma$, number of iterations $T$, mean update size $\mu = \lambda/2$

2: **Output:** Optimal mean $\mathbf{m}$, all seen states $\mathbf{Z}$

3: Set evolution paths: $\mathbf{p}_c \leftarrow \mathbf{0}, \mathbf{p}_\sigma \leftarrow \mathbf{0}$

4: Initialize state memory: $\mathbf{Z} \leftarrow []$

5: **for** $t = 1$ **to** $T$ **do**

6:     $\mathbf{Y} \leftarrow [], \mathbf{V} \leftarrow []$

7:     **for** $i = 1$ **to** $\lambda$ **do**

8:         Sample $z_i \sim \mathcal{N}(\mathbf{0}, \mathbf{I})$

9:         Generate offspring: $\mathbf{y}_i \leftarrow \mathbf{m} + \sigma \mathbf{B} \mathbf{D} \mathbf{z}_i$

10:         Evaluate offspring: $\mathbf{v}_i \leftarrow f(\mathbf{y}_i)$

11:         Save offspring: $\mathbf{Y} \leftarrow \mathbf{y}_i, \mathbf{V} \leftarrow \mathbf{v}_i$

12:     **end for**

13:     $\mathbf{Z} \leftarrow \text{sort}(\text{zip}((\mathbf{Y}, \mathbf{V})), \text{key=lambda k: k}[1])$

14:     $\mathbf{Y} \leftarrow Z[:, 0]$

15:     Calculate mean: $\mathbf{m}' \leftarrow m + \sum_{i=1}^{\mu} w_i(y_{i:\lambda} - m)$

16:     Update evolution paths: $\mathbf{p}_c \leftarrow (1 - c_{\text{cov}})\mathbf{p}_c + \sqrt{c_{\text{cov}}(2 - c_{\text{cov}})\mu_{\text{eff}}}\mathbf{B}^{-1}(\mathbf{m} - \mathbf{m}')/\sigma$

17:     $\mathbf{p}_\sigma \leftarrow (1 - c_\sigma)\mathbf{p}_\sigma + \sqrt{c_\sigma(2 - c_\sigma)\mu_{\text{eff}}}\mathbf{B}^{-1}(\mathbf{m} - \mathbf{m}')/\sigma$

18:     Update covariance matrix: $\mathbf{C} \leftarrow (1 - c_1 - c_\mu)\mathbf{C} + c_1 \mathbf{p}_c \mathbf{p}_c^T + c_\mu \sum_{i=1}^{\mu} w_i \mathbf{y}_i \mathbf{y}_i^T$

19:     Update step size: $\sigma \leftarrow \sigma \exp\left(\frac{c_\sigma}{d_\sigma}\left(\frac{\|\mathbf{p}_\sigma\|}{E[\|z\|]} - 1\right)\right)$

20:     Update mean: $\mathbf{m} \leftarrow \mathbf{m}'$

21: **end for**

22: **return** $\mathbf{m}, \mathbf{Z}$

---

For reference in Algorithm 2, $\mathbf{B}$ is the covariance matrix adaptation matrix, $\mathbf{D}$ is the diagonal matrix that scales the random sample $z_i$, $c_{\text{cov}}$ controls the adaptation of the covariance matrix, $c_\sigma$ controls the adaptation of the step size, $c_1$ is the learning rate for the rank-one update of the covariance matrix, $c_\mu$ is the learning rate for the rank-$\mu$ update of the covariance matrix, $d_\sigma$ is the damping parameter for the step size adaptation, and $\mu_{\text{eff}}$ is the effective population size, calculated as $\frac{1}{\sum_{i=1}^{\lambda} w_i^2}$, where $w_i$ are selection weights with $w_1 \geq w_2 \geq \cdots \geq w_\mu$ and $\mu \leq \lambda/2$ which are set by default following Hansen (2006) (equation 49-53). These parameters are internal to the CMA-ES algorithm and are typically set based on empirical observations or tuning for specific optimization problems. For more details please see Hansen (2006). We note for all

CMA-ES instances EvoSBDD uses 5 iterations with a pop size of 20. We use the identity matrix for the initial covariance matrix and initial step size of 0.25 and 1 for the GNN and LLM latent spaces respectively

**Controlled Generation**   Controlled molecule generation enables the use of small perturbations to the latent representations of off-the-shelf molecular autoencoders for property-guided optimization (Gao et al., 2022). Typical workflows require a seed molecule as the starting point for the optimization. In the context of CMA-ES, the seed molecule is encoded and set as the initial mean of the CMA-ES distribution, which is used to learn a more optimal distribution directly in the latent space ( Algorithm 2). In contrast to traditional seed-based optimization, EvoSBDD performs best when replacing the seed molecule *entirely* with scaled Gaussian noise. EvoSBDD avoids the pitfalls of low diversity and inefficiency as it can avoid starting with a seed molecule entirely and iteratively make small changes on a random noise initialization to gradually improve the molecular properties due to inherent latent structure (*i.e.* more smooth or clustered latent representations).

We also experimented with the traditional seed-based optimization and integrated a noised encoding scheme to prevent overfitting and bias to the reference molecule properties. The addition of initial noise improves most benchmarks compared to using the reference molecule from the test set. The amount of noise was selected to ensure a 0% reconstruction rate from the autoencoding process to prevent the optimization from ever seeing the reference molecule.

## C. Oracle Function Design

### C.1. AutoDock Vina Redocking

AutoDock Vina (ADV) (Eberhardt et al., 2021) typically starts by placing the ligand in various initial positions and orientations around the binding site on the protein. These initial placements are sampled randomly or based on some heuristic. This component is key since by not caring about the initial positions we can save a lot of time by doing the entire molecule generation in 2D space.

Once the initial placements are made, ADV performs global and local optimizations to refine the ligand conformation. ADV utilizes a genetic algorithm for the global search of the conformational space. The genetic algorithm generates a population of potential ligand conformations, evaluates their binding affinities using the scoring function, and evolves the population over multiple generations. This allows ADV to explore a wide range of ligand conformations. After the global search, the algorithm employs BFGS local optimization using a gradient-based algorithm (quasi-Newton method) to refine the most promising ligand conformations. This step aims to improve the accuracy of the predicted binding modes. Through this, ADV explores different torsional angles and atomic positions to minimize the overall energy of the system. After local optimization, ADV selects the binding pose with the lowest energy as the final result. This pose represents the algorithm's prediction for the most energetically favorable conformation of the ligand in the binding site.

By only asking ADV to generate a single mode or docked pose, we reduce the overall computational cost as it just has to return the single best result instead of generated N results with a min rmsd difference default 1 angstrom.

### C.2. AutoDock Vina Empirical Scoring Function

The scoring function in AutoDock Vina is empirical, meaning that it is derived from experimental data and knowledge about the physics and chemistry of molecular interactions. It aims to capture the contributions of various forces, such as van der Waals forces, electrostatic interactions, hydrogen bonding, and desolvation effects.

Grid-Based Energy Evaluation: Vina uses a grid-based approach to represent the interaction energy between the ligand and the protein. A 3D grid is generated around the protein, and each grid point is assigned a potential energy based on the contributions from different force fields. This grid allows for efficient evaluation of interactions at different ligand conformations.

Lennard-Jones (van der Waals) Potential: The van der Waals interaction is modeled using the Lennard-Jones potential, which describes the attractive and repulsive forces between atoms. The potential includes terms for the dispersion (attractive) and repulsion (steric) forces, which depend on the distance between atom pairs.

Coulombic (Electrostatic) Potential: The electrostatic interaction is modeled using the Coulombic potential, which describes the interaction between charged particles. In Vina, partial charges on atoms are considered, and the distance-dependent Coulombic potential accounts for both attractive and repulsive electrostatic forces.

Hydrogen Bonding: ADV includes a term to account for hydrogen bonding interactions, which are crucial in molecular recognition. Hydrogen bonds are characterized by the distance and angle between hydrogen donor and acceptor atoms. The scoring function penalizes deviations from optimal nondirectional hydrogen bond geometry.

Torsional Terms: The scoring function considers torsional terms to account for the flexibility of ligand molecules. Torsional energy reflects the strain or relief associated with rotations around flexible bonds.

Final Score: The overall score for a given ligand conformation is the sum of the individual energy contributions from the Lennard-Jones potential, Coulombic potential, hydrogen bonding, and torsional terms. The goal is to find ligand conformations that minimize the total energy and, consequently, maximize the predicted binding affinity.

## C.3. UniDock

UniDock (Yu et al., 2023) is a GPU-accelerated molecular docking program that supports various scoring functions, including vina, which achieves more than 1000-fold speed-up with high accuracy compared with the AutoDock Vina single-CPU-core version, outperforming reported GPU-accelerated docking programs including AutoDock-GPU and Vina-GPU. Uni-Dock works by dividing molecules into batches and simultaneously docking batches of molecules using hundreds of concurrent threads for each molecule.

We found UniDock overestimates scores by 0.14-0.34 kcal/mol on average when compared to the AutoDock Vina scoring function due to likely subtle differences in the weightings of the Vina scoring function. The error range spanned from -1 to +1 kcal/mol, but overall, UniDock provides more accurate binding affinity and is significantly faster than ADV. In our evaluations, all generated poses are re-scored with Vina (score only, no re-docking so the structure remains the same) to ensure a fair comparison. This way, we compare with the same scoring function as prior methods.

We note we use UniDock's default balanced mode, which has a UniDock-exhaustiveness of 384 and max step of 40 with the vina score function. We emphasize that while UniDock and ADV both the parameters such as exhaustiveness, the magnitudes are not directly comparable due to parallelization implementations. We note UniDock results have been shown to improve by using detailed mode which offers greater accuracy at the cost of longer inference time.

In comparison to DecompOpt (Zhou et al., 2024), which uses 30 rounds of optimization and samples 20 molecules in each round where the QED, SA, and Vina Min are calculated for each, EvoSBDD uses $R$ rounds of sampling 100 molecules broken into 5 BBO time steps. We emphasize that Vina Min cannot be replaced with Vina dock for time constraints and the fact that then there would be a distribution shift in the structural changes needed to achieve the resulting binding affinity and what was done by the model. The model would not learn what changes are needed to achieve strong binding affinities but which atom types ADV can use to construct the best pose, which is what EvoSBDD is doing but more efficiently.

## C.4. Oracle Setup

We use an AutoDock Vina-based and UniDock-based (with Vina scoring function) oracle functions for SBDD: Binding Affinity $= f(\text{SMILES})$. In practice, the Vina-based function is run in parallel with batch size 20, exhaustiveness 16, and max step (controls the local optimization?) set by Vina's heuristic.

As an initial 3D structure for Vina optimization, we use RDKit ETKDG without external force field relaxations.

**Vina Parameter Definitions** Exhaustiveness in AutoDock Vina controls the thoroughness of the global search. It determines the number of genetic algorithm (GA) MC runs that are performed during the optimization process. A higher exhaustiveness value leads to a more exhaustive search but requires more computational resources.

The max_steps parameter is related to the maximum number of energy evaluations per local search. It limits the number of steps that the local optimization algorithm takes for each binding mode. This parameter influences the precision of the local search but does not significantly affect the overall exhaustiveness of the global search. By default, AutoDock Vina uses a heuristic to automatically determine an appropriate value for max steps based on the size of the ligand and the number of flexible torsions. The heuristic is designed to balance the need for a sufficiently detailed search with computational efficiency.

The max steps parameter controls the maximum number of optimization steps (iterations) performed during the local search with gradient descent for each binding pose. The local optimization is a crucial step where the algorithm refines the initial placement of the ligand to find a more energetically favorable conformation.

We note that even with setting an explicit exhaustiveness, the time spent on the search is varied heuristically depending on the number of atoms, flexibility, etc. ADV expresses that "it does not make sense to spend extra time searching to reduce the probability of not finding the global minimum of the scoring function beyond what is significantly lower than the probability that the minimum is far from the native conformation"[2]. Furthermore, ADV notes that increasing exhaustiveness increases the time linearly and decreases the probability of not finding the minimum exponentially

---

[2]https://vina.scripps.edu/manual/

# D. SBDD Evaluations and Ablations

Here we present further SBDD ablations that include more EvoSBDD parameterizations and further benchmarks. We note it takes 2.8 (LLM) and 4.0 (GNN) seconds to generate 100 2D molecules for reference. We chose 8 restarts as the initial comparison as EvoSBDD with AutoDock Vina is on the same inference time cost as Pocket2Mol (Peng et al., 2022). When switching to UniDock we ran 140 restarts to match the DecompDiff (Guan et al., 2023b) time. Given that the docking score increases with more BBO restartsm we expect further improved docking at the slight cost of diversity if we let run for 9000 seconds to match recent DecomptOpt (Zhou et al., 2024). We use this serial time comparison to respect compute limited fairness but note that all EvoSBDD restarts can be run in parallel in 45 seconds.

Table 3: Full Structure-based Drug Discovery Benchmarks. The poor validity of EvoSBDD-GNN stems from failing on 8 reference molecules due to MoFlow's heavy atom encoding limit.

| | Validity (↑) | Vina Dock (↓) | | High Affinity (↑) | | QED (↑) | | SA (↑) | | Diversity (↑) | | Lipinski (↑) | Success Rate (↑) | Time (↓) |
| | Avg. | Avg. | Med. | Avg. | Med. | Avg. | Med. | Avg. | Med. | Avg. | Med. | Avg. ± Std. | Avg. | Gen + Score. |
|---|---|---|---|---|---|---|---|---|---|---|---|---|---|---|
| Reference | 100% | -7.45 | -7.26 | - | - | 0.48 | 0.47 | 0.73 | 0.74 | - | - | 4.34 ± 1.14 | 25.0% | 300 |
| **Generative** | | | | | | | | | | | | | | |
| liGAN | - | -6.33 | -6.20 | 21.1% | 11.1% | 0.39 | 0.39 | 0.59 | 0.57 | 0.66 | 0.67 | - | 3.9% | - |
| GraphBP | - | -4.80 | -4.70 | 14.2% | 6.7% | 0.43 | 0.45 | 0.49 | 0.48 | **0.79** | **0.78** | 4.83 ± 0.37 | 0.1% | 310 |
| AR | 92.95% | -6.75 | -6.62 | 37.9% | 31.0% | 0.51 | 0.50 | 0.63 | 0.63 | 0.70 | 0.70 | 4.78 ± 0.51 | 7.1% | 19959 |
| Pocket2Mol | 98.31% | -7.15 | -6.79 | 48.4% | 51.0% | 0.56 | 0.57 | 0.74 | 0.75 | 0.69 | 0.71 | 4.93 ± 0.27 | 24.4% | 2804 |
| TargetDiff | 90.35% | -7.80 | -7.91 | 58.1% | 59.1% | 0.48 | 0.48 | 0.58 | 0.58 | 0.72 | 0.71 | 4.59 ± 0.83 | 10.5% | 3728 |
| DiffSBDD | 85.01% | -8.03 | -7.74 | 55.3% | 56.6% | 0.47 | 0.47 | 0.55 | 0.56 | 0.76 | 0.76 | 4.70 ± 0.64 | 6.0% | 460 |
| DecompDiff | 71.96% | -8.39 | -8.43 | 64.4% | 71.0% | 0.45 | 0.43 | 0.61 | 0.60 | 0.68 | 0.68 | 4.29 ± 0.97 | 24.5% | 6489 |
| **Optimization** | | | | | | | | | | | | | | |
| TacoGFN | 99.27% | -7.41 | -7.50 | 58.9% | 59.0% | **0.68** | **0.72** | 0.78 | 0.79 | 0.65 | 0.65 | 4.94 ± 0.24 | 29.9% | 303 |
| TacoGFN-AL | 99.28% | -7.68 | -7.70 | 64.3% | 64.0% | 0.64 | 0.66 | **0.81** | **0.82** | 0.66 | 0.66 | 4.93 ± 0.25 | 36.6% | 303 |
| RGA | - | -8.01 | -8.17 | 64.4% | 89.3% | 0.57 | 0.57 | 0.71 | 0.73 | 0.41 | 0.41 | - | 46.2% | - |
| TargetDiff+Opt (ICLR24) | - | -8.30 | -8.15 | 71.5% | 95.9% | 0.66 | 0.68 | 0.68 | 0.67 | 0.31 | 0.30 | - | 25.8% | ¿3728 |
| DecompOpt (ICLR24) | - | -8.98 | -9.01 | 73.5% | 93.3% | 0.48 | 0.45 | 0.65 | 0.65 | 0.60 | 0.61 | | 52.5% | 9241 |
| **Ours** | | | | | | | | | | | | | | |
| EvoSBDD-GNN (8R Vina) | 92% | -7.82 | -7.75 | **85.6%** | **100%** | 0.48 | 0.49 | 0.61 | 0.61 | 0.61 | 0.63 | 4.60 ± 0.91 | 20.9% | 3272 |
| EvoSBDD (8R Vina) | **100%** | -8.66 | -8.62 | **85.6%** | **100%** | 0.55 | 0.57 | 0.75 | 0.77 | 0.65 | 0.66 | 4.70 ± 0.75 | 50.4% | 2346 |
| EvoSBDD (8R) | **100%** | **-9.01** | **-9.03** | **86.1%** | **100%** | 0.57 | 0.59 | 0.75 | 0.78 | 0.65 | 0.66 | 4.75 ± 0.65 | 59.3% | 360 |
| **EvoSBDD** ($\alpha = 0$, $\sigma = 1$, 8R) | **100%** | **-9.09** | **-9.20** | 82.1% | **100%** | 0.65 | 0.67 | 0.78 | 0.79 | 0.65 | 0.66 | **4.96 ± 0.21** | **73.5%** | 360 |
| EvoSBDD (46R) | **100%** | **-9.75** | **-9.75** | 95.4% | **100%** | 0.54 | 0.55 | 0.75 | 0.77 | 0.63 | 0.63 | 4.75 ± 0.65 | 69.2% | 2520 |
| EvoSBDD (140R) | **100%** | **-10.03** | **-10.06** | 97.8% | **100%** | 0.53 | 0.53 | 0.75 | 0.76 | 0.62 | 0.63 | 4.76 ± 0.63 | 72.3% | 6300 |
| **EvoSBDD** (140R+Noise) | **100%** | **-10.27** | **-10.36** | 96.5% | **100%** | 0.53 | 0.52 | 0.75 | 0.77 | 0.63 | 0.63 | 4.84 ± 0.44 | **78.8%** | 6300 |
| **EvoSBDD** ($\alpha = 0$, $\sigma = 1$, 140R) | **100%** | **-10.14** | **-10.27** | 94.4% | **100%** | 0.59 | 0.59 | 0.77 | 0.77 | 0.62 | 0.62 | 4.91 ± 0.29 | **86.4%** | 6300 |

Tab. 3 demonstrates the full EvoSBDD results on the CrossDocked2020 benchmarks, including more runs at various restart amounts, Vina vs UniDock usage, and GNN vs the default LLM latent space. Here we see there is a trade-off in the optimization objective. We note the GNN latent space has a heavy atom encoding limit leading to 8/100 failures of the reference model. As we run more independent BBO restarts, we see that the docking results improve at the cost of diversity. Further, we see that diversity and docking are greatly increased by starting from random noise rather than the provided reference molecules.

We stress that as the existing central metric for SBDD accuracy is the average redocking score over 10,000 protein-ligand complexes (100 protein pockets with 100 sampled ligands each), it is crucial to ensure all methods are compared over the same sample size. We used the published generated molecules for TargetDiff, Pocket2Mol, AR, DiffSBDD, and DecompDiff for this evaluation. We note TacoGFN already normalizes its results and zeros out the docking score for its failed molecules yielding a fair value thus not included here. Tab. 2 demonstrates that validity can largely impact results and that being able to just generate a single great ligand is not a plausible design goal for generative ML methods. This evaluation can be extended to more recent methods pending the release of their code/generated molecules

Table 4: Ring Size breakdown of the generated molecules.

| Ring Size | Ref. | liGAN | AR | Pocket2Mol | TargetDiff | EvoSBDD (8R, Vina) | EvoSBDD (46R, Vina) | EvoSBDD-GNN |
|---|---|---|---|---|---|---|---|---|
| 3 | 1.7% | 28.1% | 29.9% | 0.1% | 0.0% | 1.6% | 1.6% | 9.3% |
| 4 | 0.0% | 15.7% | 0.0% | 0.0% | 2.8% | 2.6% | 2.5% | 5.1% |
| 5 | 30.2% | 29.8% | 16.0% | 16.4% | 30.8% | 31.8% | 31.7% | 30.3% |
| 6 | 67.4% | 22.7% | 51.2% | 80.4% | 50.7% | 55.6% | 54.6% | 44.2% |
| 7 | 0.7% | 2.6% | 1.7% | 2.6% | 12.1% | 6.4% | 7.4% | 6.5% |
| 8 | 0.0% | 0.8% | 0.7% | 0.3% | 2.7% | 1.7% | 1.7% | 2.3% |
| 9 | 0.0% | 0.3% | 0.5% | 0.1% | 0.9% | 0.3% | 0.5% | 2.0% |

Tab. 4 demonstrates the percentage breakdown of the generated molecules ring size as done in Guan et al. (2023a). We see that EvoSBDD enables a molecule distribution that more closely resembles the reference test set. We found no significant changes in the EvoSBDD ring distribution (¡ 0.1%) when we increased restarts, introduced active noised encoding, or removed the reference molecule entirely.

Table 5: Black Box Ablations: Here, we vary the population size (ps), the number of iterations (iter), and the total number of restarts (R). Note * denotes UniDock raw score compared to parallelized AutoDock Vina. We provide Base, Max, and Ultimate results with Vina and UniDock scoring to demonstrate how Unidock, on average, increases success rate by 2% and docking score by 0.14-0.2 kcal/mol. Gaussian noise was added up to a scalar factor until the latent space could not reconstruct any of the 100 reference ligands. Noising $\sigma = 1.3$ improves overall performance at the cost of high affinity. We no longer start from the reference but by doing so we unblock getting stuck in local minima. This is furthered by removing all dependence on the reference molecule and starting from pure noise $\alpha = 0$, $\sigma = 1$. $\gamma$ determines the scaling factor of QED and SA for the BBO oracle function (default = 0) similar to the optimization objective in Zhou et al. (2024).

| | Validity (↑) | Vina Dock (↓) | | High Affinity (↑) | | QED (↑) | | SA (↑) | | Diversity (↑) | | Lipinski (↑) | Success Rate (↑) | Time (↓) | |
|---|---|---|---|---|---|---|---|---|---|---|---|---|---|---|---|
| | Avg. | Avg. | Med. | Avg. | Med. | Avg. | Med. | Avg. | Med. | Avg. | Med. | Avg. ± Std. | Avg. | Gen. | Gen + Score. |
| Reference | 100% | -7.45 | -7.26 | - | - | 0.48 | 0.47 | 0.73 | 0.74 | - | - | 4.34 ± 1.14 | 25.0% | - | 300 |
| EvoSBDD (ps 20, iter 5, 8R) | 100% | -9.01 | -9.03 | 86.1% | 100% | 0.57 | 0.59 | 0.75 | 0.78 | 0.65 | 0.66 | 4.75 ± 0.65 | 59.3% | 40 | 360 |
| EvoSBDD (ps 20, iter 5, 56R) | 100% | -9.75 | -9.75 | 95.4% | 100% | 0.54 | 0.55 | 0.75 | 0.77 | 0.63 | 0.63 | 4.75 ± 0.65 | 69.2% | 280 | 2520 |
| EvoSBDD (ps 20, iter 5, 140R) | **100%** | **-10.03** | **-10.06** | **97.8%** | 100% | 0.53 | 0.53 | 0.75 | 0.76 | 0.62 | 0.63 | 4.76 ± 0.63 | **72.3%** | 400 | 6300 |
| EvoSBDD ($\sigma = 1.3$, ps 20, iter 5, 140R) | **100%** | **-10.27** | **-10.36** | 96.5% | 100% | 0.53 | 0.52 | 0.75 | 0.77 | 0.63 | 0.63 | **4.84 ± 0.44** | **78.8%** | 400 | 6300 |
| **EvoSBDD** ($\alpha = 0$, $\sigma = 1$, 140R) | **100%** | **-10.14** | **-10.27** | 94.4% | **100%** | 0.59 | 0.59 | 0.77 | 0.77 | 0.62 | 0.62 | 4.91 ± 0.29 | **86.4%** | 400 | 6300 |
| EvoSBDD (ps 20, iter 5, 1R)* | 93.38% | -7.60 | -7.54 | 55.6% | 62.3% | 0.56 | 0.59 | 0.76 | 0.77 | 0.71 | 0.72 | 4.69 ± 0.76 | 31.4% | 3 | 45 |
| EvoSBDD (ps 20, iter 5, 2R)* | 99.97% | -8.31 | -8.27 | 74.8% | 100% | 0.58 | 0.61 | 0.75 | 0.77 | 0.67 | 0.68 | 4.72 ± 0.72 | 44.5% | 6 | 90 |
| EvoSBDD (ps 20, iter 5, 4R)* | 100% | -8.77 | -8.78 | 83.8% | 100% | 0.58 | 0.60 | 0.75 | 0.78 | 0.66 | 0.67 | 4.74 ± 0.70 | 55.3% | 12 | 180 |
| EvoSBDD (ps 20, iter 5, 6R)* | 100% | -9.00 | -9.01 | 87.4% | 100% | 0.57 | 0.60 | 0.75 | 0.78 | 0.65 | 0.66 | 4.74 ± 0.69 | 59.0% | 18 | 270 |
| EvoSBDD (ps 20, iter 5, 8R)* | 100% | -9.14 | -9.14 | 89.7% | 100% | 0.57 | 0.59 | 0.75 | 0.78 | 0.65 | 0.66 | 4.75 ± 0.65 | 61.6% | 23 | 360 |
| EvoSBDD ($\sigma = 1.3$, ps 20, iter 5, 8R)* | 100% | -9.34 | -9.32 | 89.0% | 100% | 0.58 | 0.59 | 0.76 | 0.77 | 0.66 | 0.66 | 4.85 ± 0.49 | 71.4% | 23 | 360 |
| EvoSBDD ($\alpha = 0$, $\sigma = 1$, ps 20, iter 5, 8R)* | 100% | -9.34 | -9.41 | 86.5% | 100% | 0.65 | 0.67 | 0.78 | 0.79 | 0.65 | 0.66 | 4.96 ± 0.21 | 77.1% | 23 | 360 |
| EvoSBDD ($\gamma = 1$, $\sigma = 1.3$, ps 20, iter 5, 8R)* | 100% | -9.34 | -9.37 | 88.4% | 100% | 0.60 | 0.60 | 0.77 | 0.78 | 0.66 | 0.67 | 4.88 ± 0.41 | 71.4% | 23 | 360 |
| EvoSBDD ($\gamma = 2$, $\sigma = 1.3$, ps 20, iter 5, 8R)* | 100% | -9.26 | -9.26 | 88.5% | 100% | 0.62 | 0.64 | 0.77 | 0.78 | 0.66 | 0.67 | 4.89 ± 0.43 | 70.3% | 23 | 360 |
| EvoSBDD ($\gamma = 5$, $\sigma = 1.3$, ps 20, iter 5, 8R)* | 100% | -9.13 | -9.16 | 86.1% | 100% | 0.64 | 0.66 | 0.77 | 0.78 | 0.67 | 0.68 | 4.91 ± 0.36 | 69.1% | 23 | 360 |
| EvoSBDD (ps 20, iter 5, 20R)* | 100% | -9.54 | -9.53 | 93.8% | 100% | 0.56 | 0.56 | 0.75 | 0.77 | 0.64 | 0.65 | 4.76 ± 0.66 | 66.9% | 58 | 900 |
| EvoSBDD (ps 20, iter 5, 56R)* | 100% | -9.92 | -9.94 | 96.9% | 100% | 0.54 | 0.55 | 0.75 | 0.77 | 0.63 | 0.63 | 4.75 ± 0.65 | 71.1% | 160 | 2520 |
| EvoSBDD (ps 20, iter 5, 140R)* | 100% | -10.23 | -10.24 | 98.5% | 100% | 0.53 | 0.53 | 0.75 | 0.76 | 0.62 | 0.63 | 4.76 ± 0.63 | 74.0% | 400 | 6300 |
| EvoSBDD ($\sigma = 1.3$, ps 20, iter 5, 140R)* | 100% | -10.49 | -10.59 | 97.4% | 100% | 0.53 | 0.52 | 0.75 | 0.77 | 0.63 | 0.63 | 4.84 ± 0.44 | 80.5% | 400 | 6300 |
| EvoSBDD ($\alpha = 0$, $\sigma = 1$, ps 20, iter 5, 140R)* | 100% | -10.48 | -10.56 | 95.6% | 100% | 0.59 | 0.59 | 0.77 | 0.77 | 0.62 | 0.62 | 4.91 ± 0.29 | 89.6% | 400 | 6300 |

Tab. 5 demonstrates various BBO ablations where we vary key CMA-ES parameters as well as EvoSBDD noise to reference ratio as well as evaluate for multi-property optimization. Here, we see that as we increase the weight of QED and SA in the BBO reward function, QED improves as a slight cost to the docking score. Here, we also see the subtle difference between UniDock and ADV scoring, which is why we re-scored the UniDock final results with the same ADV scoring function as

prior methods as shown in the top of Tab. 5 for fair comparison. We do not expect CrossDocked2020 benchmark results to change much by increasing the number of iterations, as we can just add new restarts from random initializations. We also found the pop size to play no significant role but can be tuned for specific task performance.

**CrossDocked Efficiency**    EvoSBDD uses UniDock (single A6000 gpu) unless ADV is specified. For evaluation consistency, as UniDock overestimates scores by 0.14-0.34 kcal/mol on average, the poses are re-scored with Vina (score only, no re-docking so the structure remains the same) to ensure a fair comparison. For further discussion on UniDock please see §C.3. The prior method results in Tab. 1, were calculated with their published generated molecules if available or taken from their publication. We specify the number of BBO restarts (R) for EvoSBDD and note each restart can be run in parallel for a total of 45 seconds. We report the serial time as it is representative of the most compute-constrained setting for fairness.

We emphasize that the majority of our speed-up is due to our iterative BBO design. For all prior 3D structure methods replacing ADV redocking with UniDock would reduce inference time by 250 seconds which is an improvement of 0.3% of the prior best method. Furthermore, as demonstrated in Tab. 3 we still achieve SOTA results in significantly less time without UniDock. We note that even if diffusion models wanted to leverage UniDock for sample guidance it would be too slow as it would require 500-1000 sequential calls due to the SDE solving step. In addition, ADV structural perturbations may cause a distribution shift as the geometric changes by the docking simulation would not have a gradient signal back to the diffusion dynamics. Overall, EvoSBDD offers a balance between generation speed, accuracy, and computational resources all the while maintaining strong performance on competitive SBDD benchmarks.

Table 6: Novelty, similarity, uniqueness, and diversity of SBDD generated molecules. 100% uniqueness of every EvoSBDD run can be enforced as a selection criterion of the inner BBO loop with no impact on any other metrics ($<0.001\%$)

|  | Novelty ($\uparrow$) | Similarity ($\downarrow$) | Uniqueness ($\uparrow$) | SBDD Diversity ($\uparrow$) |
|---|---|---|---|---|
| LiGAN | 100% | 0.22 | 87.28% | 0.66 |
| AR | 100% | 0.24 | 100% | 0.70 |
| Pocket2Mol | 100% | 0.26 | 100% | 0.69 |
| TargetDiff | 100% | 0.30 | 99.63% | 0.72 |
| DecompDiff | 100% | 0.34 | 99.99% | 0.68 |
| RGA | 100% | 0.37 | 96.82% | 0.41 |
| DecompOpt | 100% | 0.36 | 100% | 0.60 |
| EvoSBDD (8R) | 100% | 0.32 | 98.50% | 0.65 |
| EvoSBDD ($\alpha = 0, \sigma = 1$, 8R) | 100% | 0.23 | 99.58% | 0.65 |
| EvoSBDD (140R) | 100% | 0.31 | 98.61% | 0.62 |
| EvoSBDD ($\sigma = 1.3$, 140R) | 100% | 0.28 | 99.22% | 0.63 |
| EvoSBDD ($\alpha = 0, \sigma = 1$, 140R) | 100% | 0.24 | 99.82% | 0.62 |
| EvoSBDD ($\alpha = 0, \sigma = 1$, 140R+Uniq) | 100% | 0.24 | 100% | 0.62 |

**Generated Novelty**    We additionally test the Novelty and Similarity of generated ligands compared with the reference ligand. Novelty is defined as the ratio of generated ligands that are different from the reference ligand of the corresponding pocket in the test set. Similarity is defined as the Tanimoto Similarity between the generated ligands and the corresponding reference ligand. The results in Tab. 6 show strong EvoSBDD performance across all metrics, with CMA-ES random noise perturbations increasing performance. Furthermore, since the black-box optimization can be tailored to the specific use case, we can add uniqueness scoring filters in the inner CMA-ES loop to discard duplicates before updating the latent distribution which we found had no significant impact on docking or property scores ($<0.001\%$ decrease). Overall, EvoSBDD can generate 100% novel, unique, and valid molecules with strong diversity and low similarity from each other.

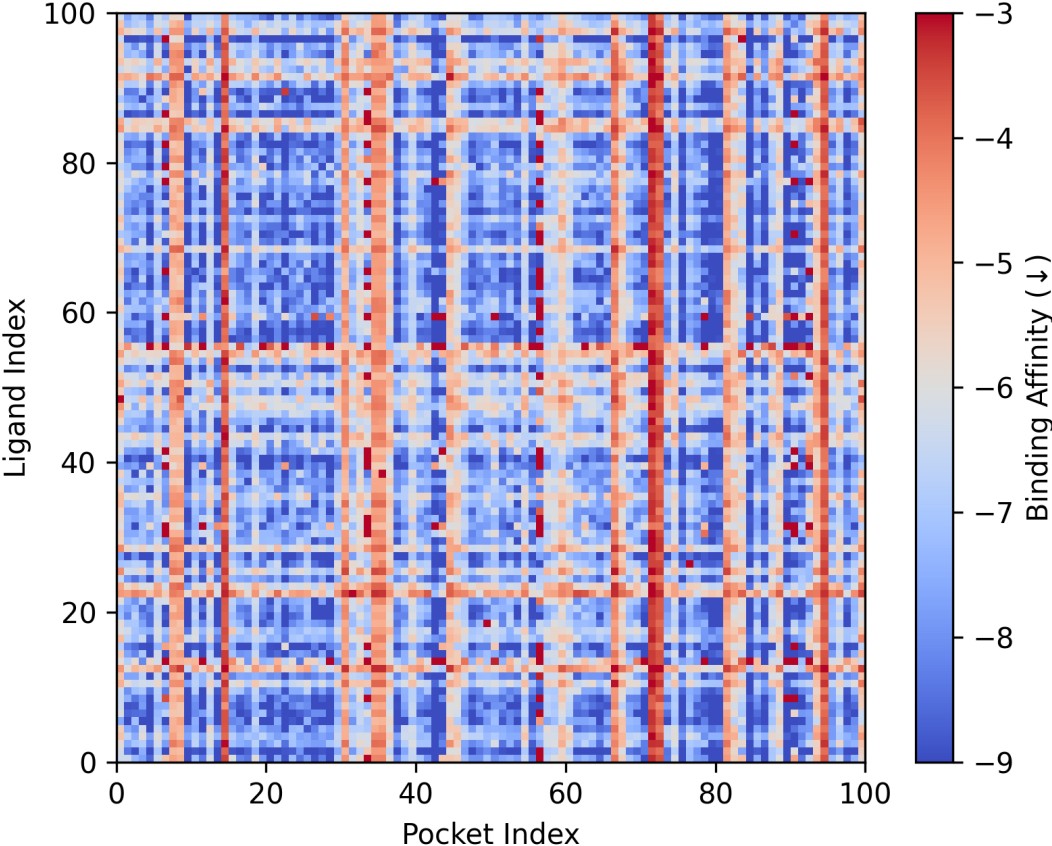

Figure 2: It is important to note that the CrossDocked2020 test set does not have high specificity. Most of the proteins exhibit strong binding with many of the reference molecules so designing only for binding affinity is not the challenge for real-world usage. To enable SBDD models to benefit drug discovery they have to take more into account the desired chemical properties and specificity.

**CrossDocked Specificity**  We note that Structure-based drug design is never about single proteins in isolation in practice. To design effective medications we have to ensure our drug molecules bind to where we intend for them to bind and not to potentially dangerous off-target sites. To gauge the specificity of the CrossDocked2020 test set we took each ligand from each of the 100 test proteins and docked it to the center of each test protein resulting in Fig. 2. Here we see that for the most part ligands are either binders or non-binders with some variational dependence per protein. This showcases the challenges of real-world drug discovery as from the ground truth data molecules do not exhibit large tendencies of specific binding. We look into novel multi-objective SBDD tasks that focus on specificity specifically in §F.

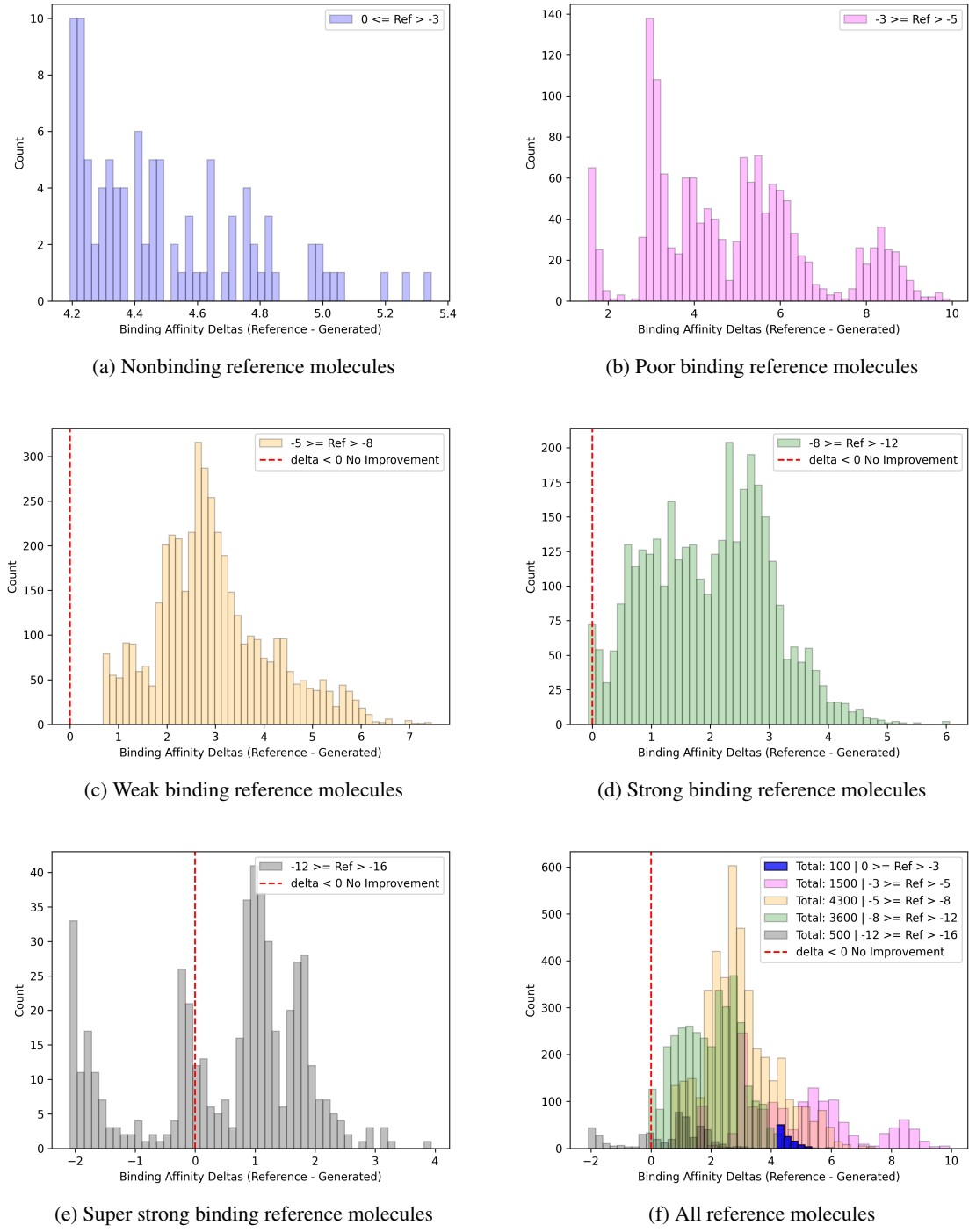

(a) Nonbinding reference molecules

(b) Poor binding reference molecules

(c) Weak binding reference molecules

(d) Strong binding reference molecules

(e) Super strong binding reference molecules

(f) All reference molecules

Figure 3: EvoSBDD-LLM Ultimate BBO correlation with reference molecule binding affinity. We see the best improvement and success when the reference molecules are weak binders. (a) distribution of docking improvement when starting from a binder with affinity ¿ -3, (b) distribution of docking improvement when starting from a binder with affinity in the range [-3, -5), (c) distribution of docking improvement when starting from a binder with affinity in the range [-5, -8), (d) distribution of docking improvement when starting from a binder with affinity in the range [-8, -12), (e) distribution of docking improvement when starting from a binder with affinity in the range [-12, -16), (f) is the superposition of all prior distributions.

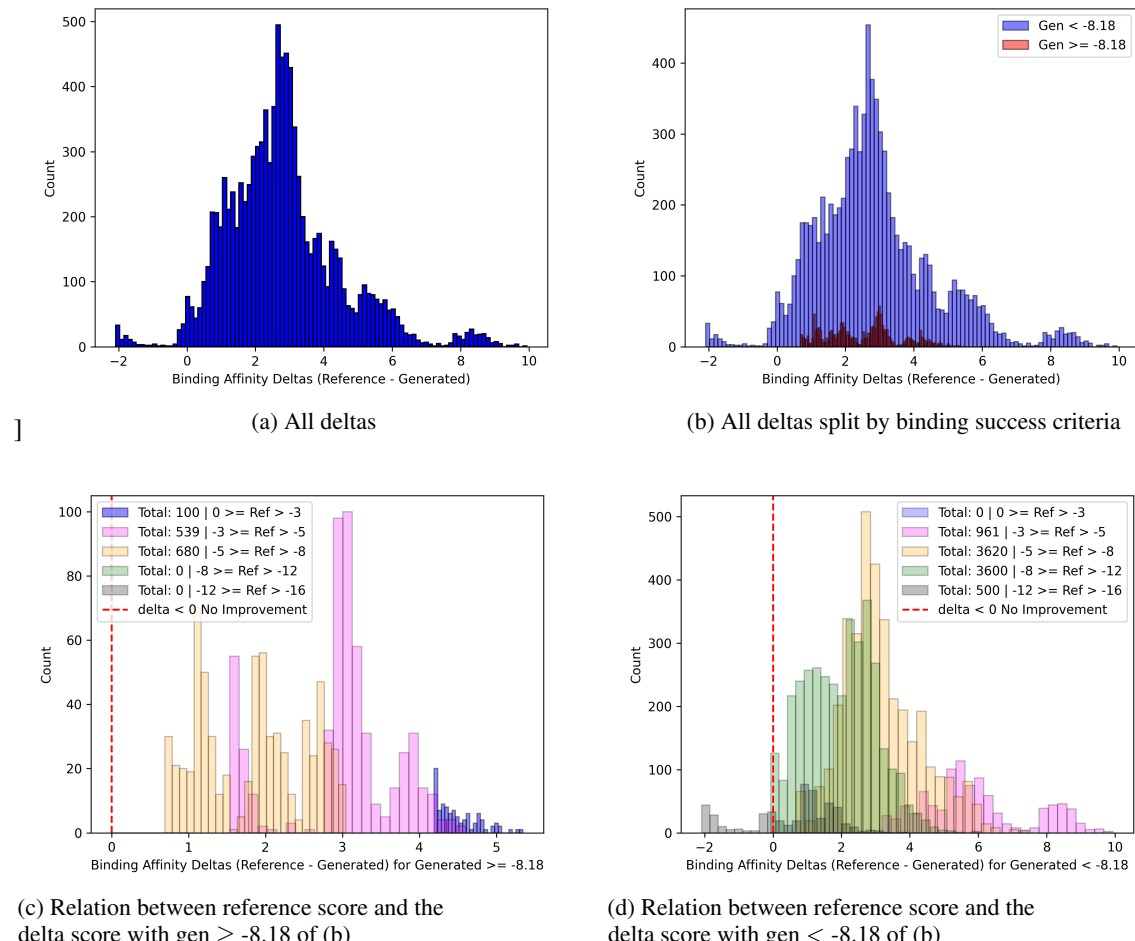

(a) All deltas

(b) All deltas split by binding success criteria

(c) Relation between reference score and the delta score with gen ≥ -8.18 of (b)

(d) Relation between reference score and the delta score with gen < -8.18 of (b)

Figure 4: Using the same deltas as Fig. 3, we demonstrate the relation to the -8.18 Vina Dock success criteria. (a) distribution of all 10,000 binding affinity differences (reference-generated), with more positive being better as binding affinities are less than zero. (b) same delta distribution but highlighted the samples that fail to hit the -8.18 kcal/mol threshold in red. (c) Zooming in on the red distribution of b, we overlay the distributions of deltas based on the reference binding affinity. (d) Zooming in on the blue distribution of b, we overlay the distributions of deltas based on the reference binding affinity.

**BBO Starting Point**   Instead of starting from random noise, which resulted in the best CrossDocked2020 results, we explore the traditional black-box optimization paradigm that requires the use of an initial starting point. Here, we proved EvoSBDD with the reference ligand to initialize the CMA-ES distribution for SBDD optimization. We see that for the most part so long as you have a decent starting point EvoSBDD can find a success as determined by §3. However, if you provide EvoSBDD with a pure random starting point we can explore more of the molecular latent space and avoid the information bottleneck resulting in improved benchmark results ( Tab. 5). Whether using the reference 2D molecule or random noise, EvoSBDD still has to encode the embedding and create a random conformer to feed to its docking oracle. We note that conformer generation and its usage in redocking-based benchmarks have also been explored in (Reidenbach & Krishnapriyan, 2023).

**Reference Dependence**   As we frame SBDD as a conditional generation problem, there is a non-negligible dependence of the generated results on the input reference ligand. Although EvoSBDD's best benchmark performance comes from not using the reference molecules but rather starting each optimization with random noise, we still conduct an extensive study on the reference conditioned results. We note that this conditional framing follows real-world lead optimization workflows but introduces an understandable reference bias that deserves further exploration.

Fig. 4 demonstrates how if the reference ligand has a binding affinity $> -3$ kcal/mol, EvoSBDD cannot generate a ligand with an affinity less than -8.18 kcal/mol, although it does show improvements of -4.2 to -5.4 kcal/mol which is still a significant improvement. We also see that in cases where the reference ligand is an extremely strong binder $< -12$ kcal/mol, it is always a success, but in a few cases, we generate worse but still strong binders of at worst -10 kcal/mol. We note that there is not a strong physical difference between -10 and -12 as further discussed in §E. We see that there is no strong correlation between reference and binding success for poor and weak binders. We see that removing the reference molecule entirely by initializing the optimization with a noised embedding such that the LLM cannot reconstruct the reference molecule improves results by a sizable margin. We hypothesize that while in a few cases, the references are good, in many cases, forcing a guaranteed poor starting point harms the optimization, whereas starting from a random point, each restart improves latent exploration and diversity, yielding a binding affinity increase of 2.3%. While this is good from an optimization perspective, the flow of information from the reference initialized procedure is more aligned with drug discovery in practice, hence our prior reference-based distribution analysis.

## E. Biological Importance

Here we will discuss what it means to generate molecules with binding affinity $< -8.18$ kcal/mol and more targeted ablations that showcase the value of efficiency in the loop optimization.

To start we provide a high-level overview of protein-ligand binding kinetics.

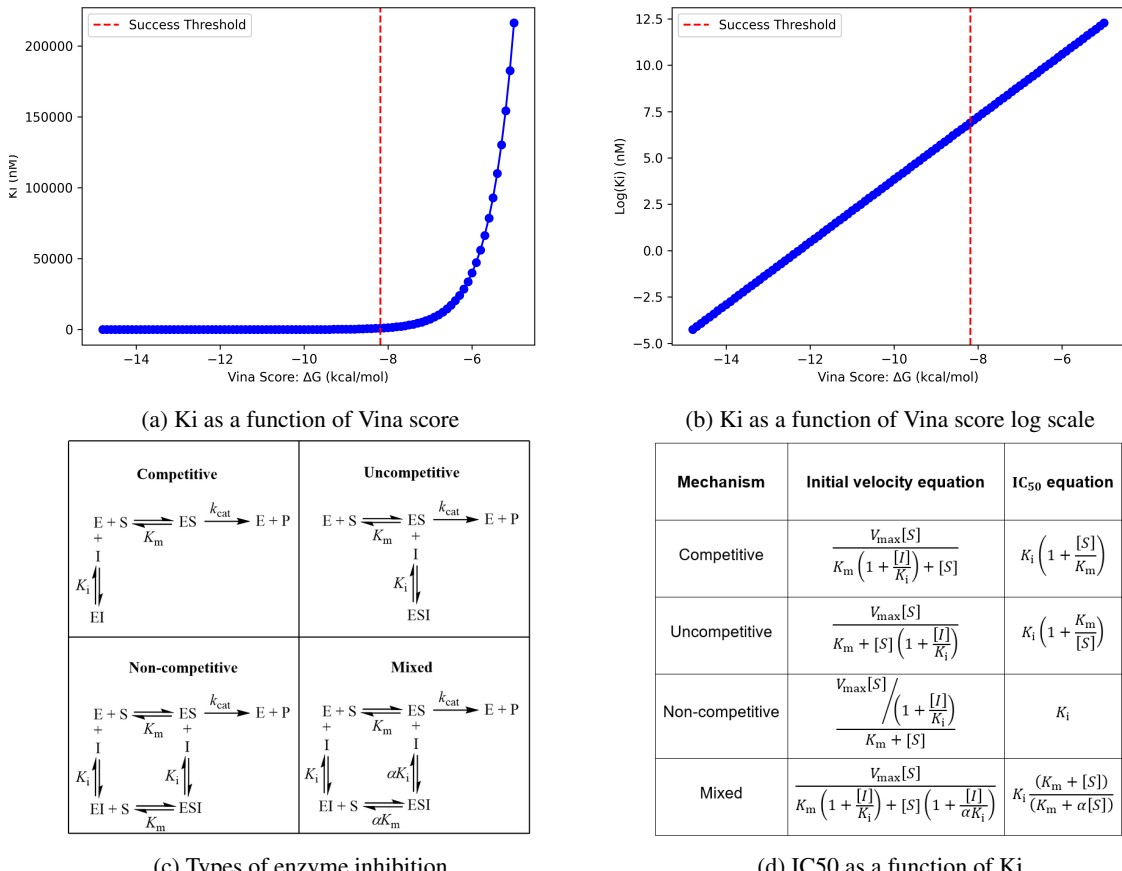

(a) Ki as a function of Vina score

(b) Ki as a function of Vina score log scale

(c) Types of enzyme inhibition

(d) IC50 as a function of Ki

Figure 5: Binding Affinity Kinetics

In biochemistry, a common medication methodology is to inhibit enzymes (proteins) that are related to the disease of interest. Several quantities, such as Ki, Kd, and IC50, are used to measure the drug potency. All of which are inversely proportional to potency.

- Ki is the equilibrium constant for the inhibition of an enzyme by a specific inhibitor. It represents the concentration of an inhibitor required to inhibit an enzyme-catalyzed reaction by 50%.

- Kd is the equilibrium constant for the dissociation of a complex, such as a protein-ligand complex. Kd represents the concentration of ligand at which half of the protein binding sites are occupied.

- IC50 is the half-maximal inhibitory concentration. It is commonly used in pharmacology to measure the potency of a drug or inhibitor. In the context of protein-ligand binding, it represents the concentration of a ligand required to inhibit a specific biological process by 50%. In the context of SBDD binding affinity $< -8.18$ kcal/mol corresponds to an IC50 of $1\mu$M.

Fig. 5 demonstrates how in the simplest uncompetitive case IC50 and Ki are identical (with others being a scalar multiple

close to 1 of Ki)[3]. Based on the thermodynamic relationship between free energy and Ki we plot the AutoDock Vina Score in units that can better measure binding effectiveness and thus drug potency. Specifically, Fig. 5 a and b show how once a ligand surpasses -8.18 kcal/mol the change in potency is quite minimal. This is important to understand as for EvoSBDD when given a strong binding reference (Fig. 3(e)), although most molecules improve upon the reference the ones that do not are still < -8.18 with very strong potency. Furthermore Fig. 5 a can be extrapolated to binding affinities > -3 kcal/mol. In this area (a single molecule in the CrossDocked test) set, EvoSBDD is unable to generate < -8.18 but does improve molecules by 5.4 kcal/mol which in potency concentrations is an increase of $110\mu M$ which is very significant.

We emphasize that as EvoSBDD performs best when replacing the input reference molecule with random Gaussian noise the same reference-based constraints are not present. We instead draw attention to the fact that once molecules reach -10kcal/mol there is little to no gain by further increasing that binding affinity. EvoSBDD is already hitting this saturation point on average and thus going forward we encourage future work to devise more realistic optimizations that cover multiple properties as improving the score to -11 while on paper may seem impressive has an exponentially small real-world impact.

## F. Further Off Target Binding Prevention Benchmarks

**Background**    At a high level, a drug that has a perfect binding affinity to its desired target is useless if it also readily binds to other dangerous sites. Unlike prior methods that blindly sample given a protein of interest, EvoSBDD can easily customize its optimization goal to encourage on-target binding while also prohibiting off-target binding, a major focus in real-world drug discovery.

Table 7: Pairwise Vina Dock scores of COX-1 and COX-2 ligand references from PDB. COX-2 Chain E is the primary binding site with binding data from BindDB (Gilson et al., 2016).

| | COX-1 Binding Affinity (↓) | COX-2 Binding Affinity (Chain G) (↓) | COX-2 Binding Affinity (Chain I) (↓) | COX-2 Binding Affinity (Chain E) (↓) |
|---|---|---|---|---|
| COX-1 Reference (Chain E) | -4.411 | -3.348 | -3.301 | -5.454 |
| COX-2 Reference (Chain G) | -4.986 | -4.919 | -4.432 | -4.565 |
| COX-2 Reference (Chain I) | -5.372 | -3.615 | -4.984 | -6.008 |
| COX-2 Reference (Chain E) | -6.012 | -5.630 | -6.777 | -8.897 |

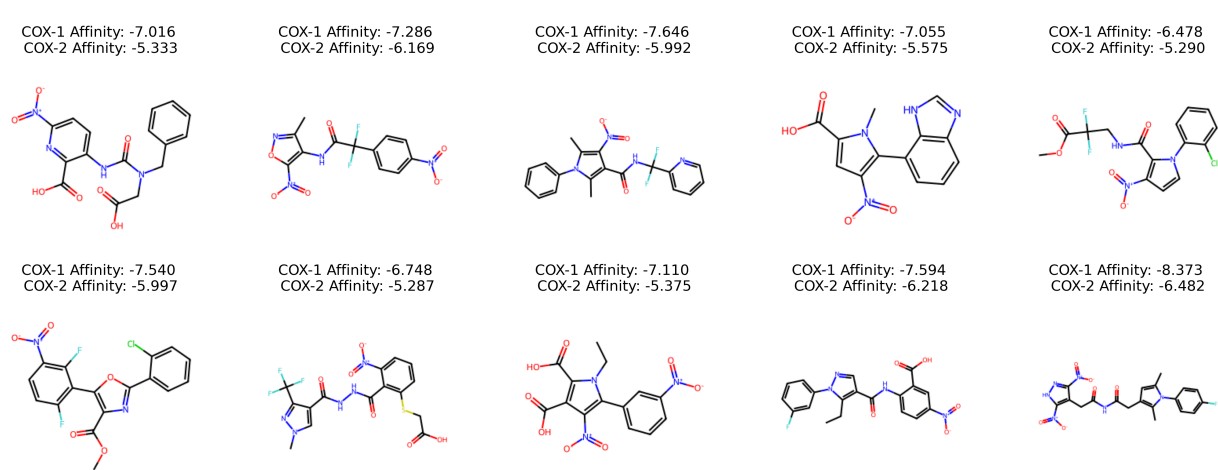

Figure 6: Top generated molecules of a single CMA-ES restart optimized to bind to COX-1 and avoid off-target binding to COX-2 (chain G). Average affinity gap -1.513 kcal/mol favoring COX-1. 10 iterations, pop size 20, using UniDock for all docking evaluations.

---

[3]Figures c and d from https://www.sciencesnail.com/science/the-difference-between-ki-kd-ic50-and-ec50-values

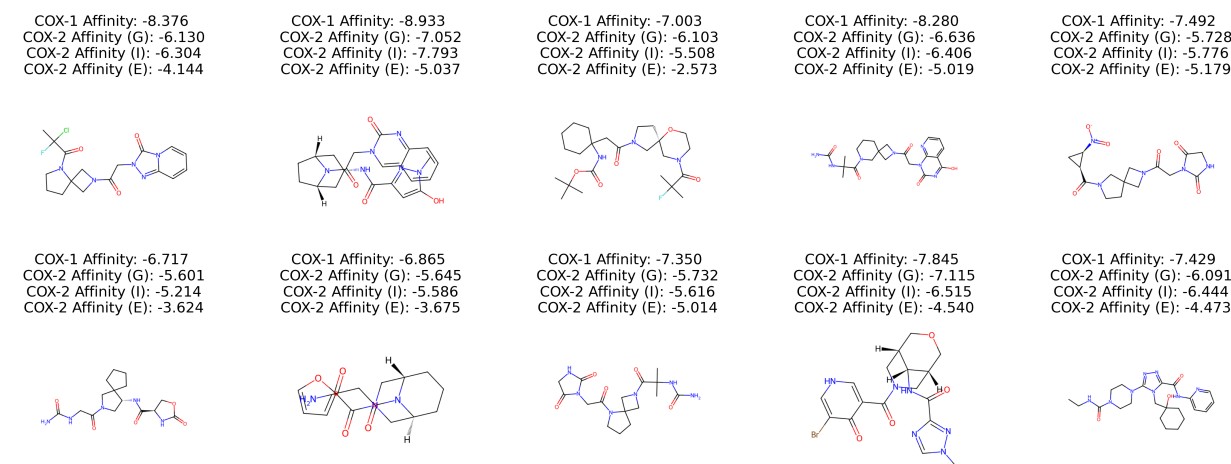

Figure 7: Top generated molecules of a single CMA-ES restart optimized to bind to COX-1 and avoid off-target binding to COX-2 (chains G, I, E). 10 iterations, pop size 20, using UniDock for all docking evaluations. Out of 200 sampled molecules, 46 have COX-1 > all COX-2 chains. This is an example of optimizing for 4 binding sites at the same time (1 on target, 3 preventative off target), which prior non-optimization methods are unable to do.

Similar to Fig. 2 we report Tab. 7 to demonstrate the pairwise binding affinities of the ligands to the native structures as found in the PDB. To demonstrate EvoSBDD's ability for specificity-driven design, we present an evaluation based on two protein isoforms, COX-1 and COX-2. COX-1 (PDB ID 6Y3C) is most known for protecting the gastrointestinal tract from stomach acids and is also involved in maintaining blood clotting for healthy kidney and platelet function (Vane et al., 1998). On the other hand, COX-2 (PDB ID 1CX2) is found at sites of inflammation (Hawkey, 2001). Furthermore, COX-2 inhibition is a major form of treatment for a variety of scenarios, including inflammation reduction for conditions like arthritis, Crohn's disease, and ulcerative colitis. Designing an inhibitor for COX-2 is fairly simple in isolation, but the simultaneous inhibition of the COX-1 isoform can lead to dangerous side effects like gastrointestinal bleeding (Brune & Patrignani, 2015). Thus, effective treatments must bind to COX-2 and not COX-1 to avoid such side effects.

**Multi-structure Target Results** To this end, we present a series of 3 evaluations, generating molecules that (1) bind to COX-1 while avoiding a single site of COX-2 (Fig. 6, (2) bind to COX-1 while avoiding all three sites of COX-2 (Fig. 7, and (3) bind to the major site (as defined by BindingDB (Gilson et al., 2016)) of COX-2 while avoiding a COX-1 (Fig. 8-Fig. 9).

We note that prior SBDD models that do not use an explicit optimization procedure cannot be used for off-target prevention. Furthermore, when compared to the compute cost of DecompOpt, EvoSBDD as it uses a single restart took 201 seconds to generate and score 200 molecules whereas DecompOpt would take roughly 18,482 and thus we are over 92x faster. For reference, if we increase the population size from 20 to 100 to generate 1000 molecules, EvoSBDD takes 320 seconds compared to DecompOpt's 92,410, thus 288x faster. We emphasize that DecompOpt's equivariant diffusion model is limited to ADV minimization optimization. Allowing the full docking simulation to change the structure during ODE/SDE diffusion sampling may create a distributional shift between the predicted structure that can be traced via gradient updates and ADV changes. This makes using UniDock with diffusion-based models for multi-target optimization not inherently straightforward.

Overall, this evaluation presents a real-world application of ML-based drug discovery workflows. In general, we see it is relatively easy to generate binders to COX-1 and COX-2 in isolation ( Tab. 7), but it is extremely difficult to generate ligands with a significant specificity for one over the other. We emphasize that prior SBDD methods, especially diffusion models that are built to be conditioned on a single 3D structural protein pocket, do not possess an efficient way to evaluate the presented multi-target benchmarks out of the box.

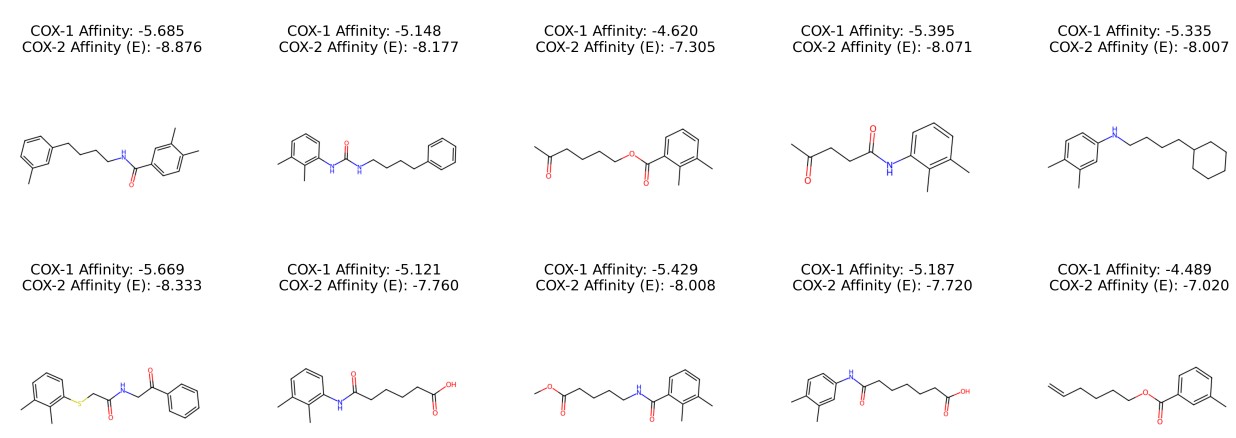

Figure 8: Top UniDock scores of generated molecules of a single CMA-ES restart optimized to bind to the main binding site of COX-2 while avoiding COX-1.

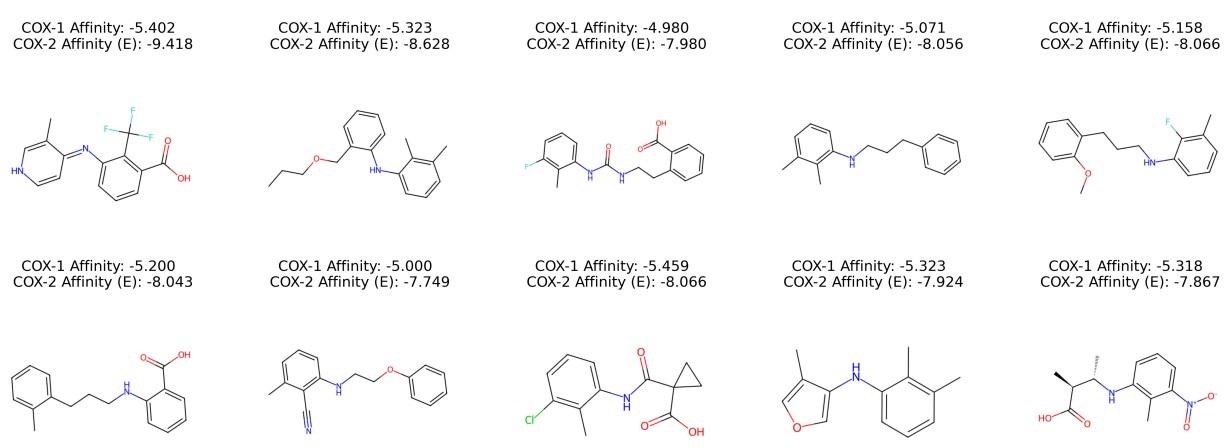

Figure 9: Top UniDock scores of generated molecules of a 10 CMA-ES restarts optimized to bind to the main binding site of COX-2 while avoiding COX-1. Notice how more restarts lead to greater diversity in top structures.

## G. PoseCheck Evaluations

We illustrate a subset of the PoseCheck (Harris et al., 2023) benchmarks that focus on the generative accuracy of the final redocked molecules. **Steric clashes** occur when the pairwise distance between a protein and ligand atom falls below the sum of their van der Waals radii. The **strain energy** is the difference between the internal energy of a relaxed and generated pose. We also compare across 4 interaction types: Hydrogen bonds (HB) occur between a hydrogen atom that is bonded to a highly electronegative atom, with the directionality of the interaction determining whether a species is a **HB Donor** or a **HB Acceptor**. **Van der Waals contacts** are weak interactions that occur between non-bonded atoms. **Hydrophobic interactions** are non-covalent interactions between non-polar molecule regions or in an aqueous environment. Fig. 10 demonstrates strong EvoSBDD performance compared to prior structure-based diffusion models, especially in limiting the average number of steric clashes(↓) to 7.89 vs 9.19 and strain energy(↓) 6.76e6 vs 1.12e9 kcal/mol when comparing to next closest method.

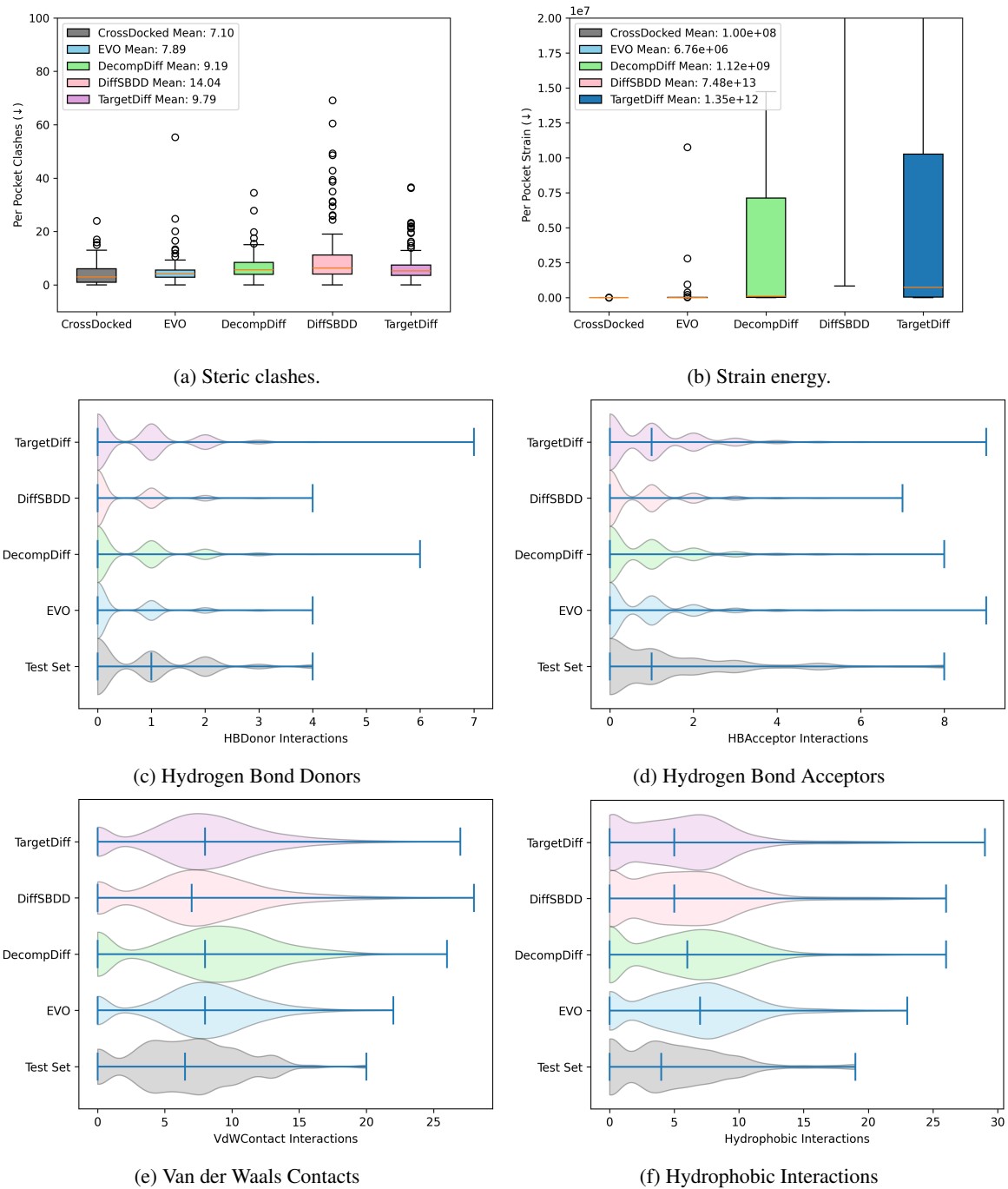

(a) Steric clashes.

(b) Strain energy.

(c) Hydrogen Bond Donors

(d) Hydrogen Bond Acceptors

(e) Van der Waals Contacts

(f) Hydrophobic Interactions

Figure 10: PoseCheck evaluations on re-docked molecules.

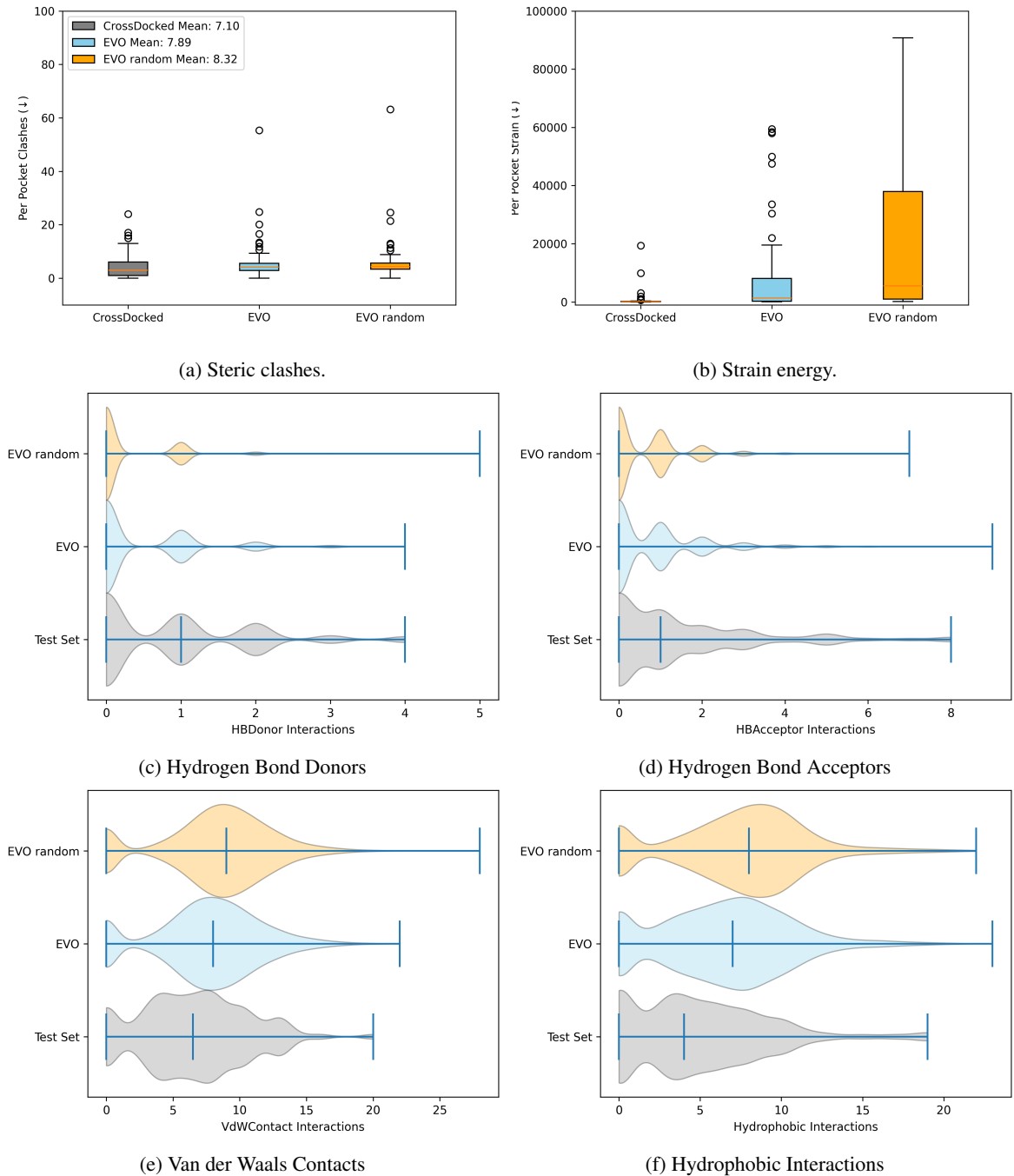

(a) Steric clashes.

(b) Strain energy.

(c) Hydrogen Bond Donors

(d) Hydrogen Bond Acceptors

(e) Van der Waals Contacts

(f) Hydrophobic Interactions

Figure 11: PoseCheck evaluations on re-docked molecules.

Here in Fig. 10 and Fig. 11, we demonstrate a subset of PoseCheck performance focusing on redocked molecule performance for EvoSBDD (referred to as EVO in the figures) when it uses the given reference molecule as well as starting from pure noise.

We see that compared to prior diffusion models, EvoSBDD generates molecules with less steric clashes and strain energy. EvoSBDD also exhibits competitive performance on all four interaction types. We note that as a majority of PoseCheck focuses on the accuracy of the non-redocked molecule, we could only evaluate EvoSBDD on the select subset. In many domain applications redocking with ADV or more principled methods or molecular dynamics simulations is relied on.

Therefore building efficient optimization methods that can leverage such redocking tools for accurate SBDD is an important goal.

