# OpenReview forum: "EvoSBDD: Latent Evolution for Accurate and Efficient Structure-Based Drug Design"
_ICML.cc/2024/Workshop/ML4LMS — ML4LMS Poster_

### Official Review · Reviewer_VDHR · 2024-06-05
**EvoSBDD**

**Rating:** 9
**Confidence:** 5

**Review:**

This is a well written paper and has proved higher performance metrics to generate valid docked structures. Specifically it uses the 1D information and provides a user-friendly approach.

---

### Official Review · Reviewer_G41v · 2024-06-11
**A target-based molecule generation pipeline is thoroughly evaluated on the CrossDocked2020 benchmark, demonstrating promising results in generation efficiency and the binding efficacy of generated molecules against selected targets.**

**Rating:** 7
**Confidence:** 5

**Review:**

This study proposes a pipeline to optimize the generation of accurately docked molecules by reframing Structure-Based Drug Design (SBDD) as a 1D-controlled latent generation problem. Instead of relying on 3D structures, this study introduce EvoSBDD, which employs a simple evolutionary algorithm in a pre-trained 1D latent space, utilizing an AutoDock Vina re-docking oracle.

The pipeline is explained thoroughly and explicitly, with clear explanations of the model components. The inclusion of a model figure and pseudocode enhances understanding. The proposed system is straightforward and easy to grasp, which adds to its appeal. Using pre-trained molecular models is a valuable addition, enabling the system to comprehend molecular structures independently of the target-compound relationship, thus allowing the design of valid and drug-like molecules. Incorporating AutoDock Vina into the design process is a good idea, providing physics-based feedback. This is a widely accepted practice in the literature and a crucial tool, as deep learning-based docking tools are not as effective according to recent studies [1].

The ablation study and experimental comparison are comprehensive. Results indicate that EvoSBDD outperforms nearly all models across most metrics. EvoSBDD demonstrates high performance on Lipinski's rule of five and average docking affinity, which are significant. However, additional metrics such as Veber's rule [2] and the pan-assay interference compounds (PAINS) filter [3] could further support the claim of drug-like molecules.

Similarity comparisons of the molecules were done in bulk, yielding a single similarity value. However, providing this similarity for each protein's corresponding generated molecules would offer more insight. This would help understand how each molecule cluster compares to real inhibitors of the given proteins. Adding a t-SNE plot could also illustrate how the generated molecules cluster in 2D planar space.

Another useful experiment would be molecular dynamics simulations for the top-5 or top-3 selected molecules to demonstrate their effectiveness in interacting with the selected proteins. While docking poses suggest potential interactions, they are static. Protein-compound interactions in biological environments are dynamic, and molecular dynamics is the best method in hand to approximate this dynamism. Confirming interactions through molecular dynamics would provide stronger evidence.

Overall, this paper is a thoroughly executed study reporting promising results in target-based molecule generation. The experiments are well-defined, and the discussion is insightful.

[1] Buttenschoen, M., Morris, G. M., & Deane, C. M. (2024). PoseBusters: AI-based docking methods fail to generate physically valid poses or generalise to novel sequences. Chemical Science.

[2] Veber, D. F. et al. Molecular properties that influence the oral bioavailability of drug candidates. J. Med. Chem. 45, 2615–2623 (2002).

[3] Baell, J. B. & Holloway, G. A. New substructure filters for removal of pan assay interference compounds (PAINS) from screening libraries and for their exclusion in bioassays. J. Med. Chem. 53, 2719–2740 (2010).

---

### Official Review · Reviewer_PkKL · 2024-06-12
**EvoSBDD in a 1D optimization method for generative models of molecules able to optimize generation through a GA and an oracle**

**Rating:** 7
**Confidence:** 3

**Review:**

**Strengths**: The method leverages pretrained latent spaces of 1D SMILES embeddings, bypassing the need for 3D coordinates and protein information during generation. This drastically speeds up the molecule generation process. The use of pretrained autoencoders ensures that generated molecules are chemically valid and meaningful, which is critical in drug discovery. The system can incorporate various desired chemical properties and protein targets into the oracle function, allowing for tailored molecule generation. EvoSBDD shows high validity (100%) and success rate (up to 86.4%), indicating robust molecule generation and optimization. EvoSBDD achieves state-of-the-art results in binding affinity, high affinity, Lipinski rules, and success rate, demonstrating its efficacy compared to other methods. The method is fully tested throughout several metrics, benchmarks and situations with already top-notch metrics.

**Weaknesses**: It would be interesting to see different genetic algorithms (there are several option with little impact on running time) and their effect on the generation. Even though the metrics reported are very competitive, more astringent thresholds on QED, binding affinity and SA should be placed, as in real DD scenarios highest quality molecular profiles are needed to advance compounds forward.